# A reduction in voluntary physical activity in early pregnancy in mice is mediated by prolactin

**Sharon R Ladyman[1,2]\*, Kirsten M Carter[1], Matt L Gillett[1], Zin Khant Aung[1], David R Grattan[1,2]\***

[1]Centre for Neuroendocrinology and Department of Anatomy, School of Biomedical Sciences, University of Otago, Dunedin, New Zealand; [2]Maurice Wilkins Centre for Molecular Biodiscovery, Auckland, New Zealand

**Abstract** As part of the maternal adaptations to pregnancy, mice show a rapid, profound reduction in voluntary running wheel activity (RWA) as soon as pregnancy is achieved. Here, we evaluate the hypothesis that prolactin, one of the first hormones to change secretion pattern following mating, is involved in driving this suppression of physical activity levels during pregnancy. We show that prolactin can acutely suppress RWA in non-pregnant female mice, and that conditional deletion of prolactin receptors (Prlr) from either most forebrain neurons or from GABA neurons prevented the early pregnancy-induced suppression of RWA. Deletion of Prlr specifically from the medial preoptic area, a brain region associated with multiple homeostatic and behavioral roles including parental behavior, completely abolished the early pregnancy-induced suppression of RWA. As pregnancy progresses, prolactin action continues to contribute to the further suppression of RWA, although it is not the only factor involved. Our data demonstrate a key role for prolactin in suppressing voluntary physical activity during early pregnancy, highlighting a novel biological basis for reduced physical activity in pregnancy.

**\*For correspondence:**
sharon.ladyman@otago.ac.nz (SRL);
dave.grattan@otago.ac.nz (DRG)

**Competing interest:** The authors declare that no competing interests exist.

## Introduction

Pregnancy and lactation represent profound physiological challenges. The sustained changes in metabolic rate typical of pregnancy have been described as 'at the limits of human physical capability,' similar to that expended in extreme physical activity such as an ultramarathon, but over a longer timeframe (*Thurber et al., 2019*). To enable the pregnant female to cope with these demands, hormonal changes associated with pregnancy drive a wide range of adaptations to maternal physiology, and in particular, complex changes in metabolic function (*Napso et al., 2018*). Using a mouse model to characterize metabolic adaptations in pregnancy, we have shown that along with increases in energy intake, pregnant females rapidly lower their energy expenditure and physical activity levels, as measured by daily voluntary running wheel activity (RWA; *Ladyman et al., 2018a*). This profound change in behavior is likely important to offset the prolonged elevation in basal metabolic rate characteristic of pregnancy (*Ladyman et al., 2018a*; *Forsum and Lof, 2007*; *Morrison, 1956*). Remarkably, however, the reduction in RWA starts very early in pregnancy before there is any significant increase in metabolic load, even before implantation (*Ladyman et al., 2018a*; *Slonaker, 1925*), and thus, can be considered an anticipatory adaptation in preparation for the future metabolic demands. Based on the very rapid change in behavior, we hypothesized that this reduction in physical activity must be mediated by the hormonal changes associated with the maternal recognition of pregnancy. In rodents, one of the very first changes in hormones in pregnancy is the mating-induced initiation of twice-daily prolactin surges that are required to maintain corpus luteum function in the ovary to

sustain the pregnancy (*Phillipps et al., 2020*). Here, we have investigated if there is a role for prolactin in mediating the pregnancy-induced suppression of RWA.

## Results

### Reduced physical activity during early pregnancy

The presence of a running wheel enables a robust assessment of an animal's choice to voluntarily engage in exercise or not. *Figure 1A* depicts daily RWA from a single representative female mouse during different reproductive states, showing the cyclical running patterns characteristic of the estrous cycle with increased running preceding each ovulation; profound reductions in RWA during pregnancy and lactation, apart from a brief increase in RWA the night after birth; and the rapid return to non-pregnant levels of RWA following weaning of pups. To enable experiments both within behavioral phenotyping cages and within their normal home cages, we investigated voluntary RWA during pregnancy independently using two different types of running wheels. Our metabolic phenotyping cages (Promethion, Sable Systems International) had a traditional upright wheel, while a saucer/low-profile wheel was used in the home cages. The upright wheel potentially takes more effort to run on, whereas the saucer wheel has less resistance and the possibility of 'coasting' on the wheel may lead to higher RWA being detected (as suggested by the higher average 'distance' measured in the saucer group vs. the upright wheel group, *Figure 1B*, $p<0.0001$, $t = 4.608$, $df = 60$) (*De Bono et al., 2006*; *Girard et al., 2001*; *Manzanares et al., 2018*). Regardless of the different equipment, all mice showed a rapid reduction in RWA early in pregnancy (*Figure 1C*, significant effect of time $p<0.0001$, $F_{(21, 1163)} = 1163$ and *Figure 1D*, significant effect of physiological state [non-pregnant, non-lactating control, henceforth referred to as 'non-pregnant' group] vs. early pregnancy: $p<0.0001$, $F_{(1, 47)} = 133.8$, Sidak's multiple comparisons; non-pregnant vs. early pregnancy: upright wheel $p<0.0001$, $t = 7.887$, $df = 47$; saucer wheel $p<0.0001$, $t = 8.461$, $df = 47$), demonstrating that this is a robust behavioral change induced as early as the first day of pregnancy. Day of birth was deemed the day pups were first observed in the home cage (checked between 0800 hr and 1700 hr). During the dark phase following the first observation of pups, RWA was significantly increased compared to either the dark phase immediately before the day of birth or the second dark phase after pups were born/observed (significant effect of time $p<0.0001$, $F_{(2, 110)} = 43.10$, Sidak's multiple comparisons: day before birth vs. day of birth $p<0.0001$, $t = 8.671$, $DF = 110.0$; day of birth vs. day after birth $p<0.0001$, $t = 7.210$, $DF = 110.0$; *Figure 1E*). This increase was observed in 88% of mice (50 out of 57) and is likely to be associated with the hormonal changes that drive the postpartum ovulation in mice, although this was not further investigated.

To determine if the pregnancy effect on physical activity was specific to RWA, we also examined general activity levels and energy expenditure in female C57BL/6J mice *without access to running wheels*. Mice were housed in Promethion metabolic and behavioral monitoring cages before and during pregnancy (as previously described in *Ladyman et al., 2018a*). We observed that pregnancy significantly influenced both energy expenditure (*Figure 1E*, repeated measures ANOVA $p=0.0002$, $F_{(1.457, 7.286)} = 39.86$) and total daily ambulation (*Figure 1F*, repeated measures ANOVA $p=0.005$, $F_{(2.063, 8.253)} = 10.24$). Both energy expenditure (*Figure 1F*, paired t-test $p=0.03$, $t = 2.812$, $df = 5$) and ambulation (*Figure 1G* paired t-test $p=0.03$, $t = 3.121$, $df = 4$) were significantly reduced in early pregnancy compared to the non-pregnant state. This suggests that early pregnancy is associated with reduced physical activity, independent of the actual form of physical activity measured. When mice are housed with running wheels, this reduction in physical activity is only detected as a major reduction in RWA, likely because this accounts for by far the greatest proportion of their overall physical activity levels (*Ladyman et al., 2018a*), whereas in the current study, when housed without running wheels, the reduction in physical activity is detected in home cage ambulation, which is the predominant form of physical activity in this situation.

### Acute effects of prolactin on physical activity and behavior

The immediate change in behavior following the onset of pregnancy suggested that it was induced by hormonal changes very early in pregnancy. We hypothesized that prolactin was the most likely candidate due to the rapid induction of twice-daily prolactin surges induced by mating in rodents. This idea was supported by the observation that the reduction in RWA persisted throughout pregnancy and

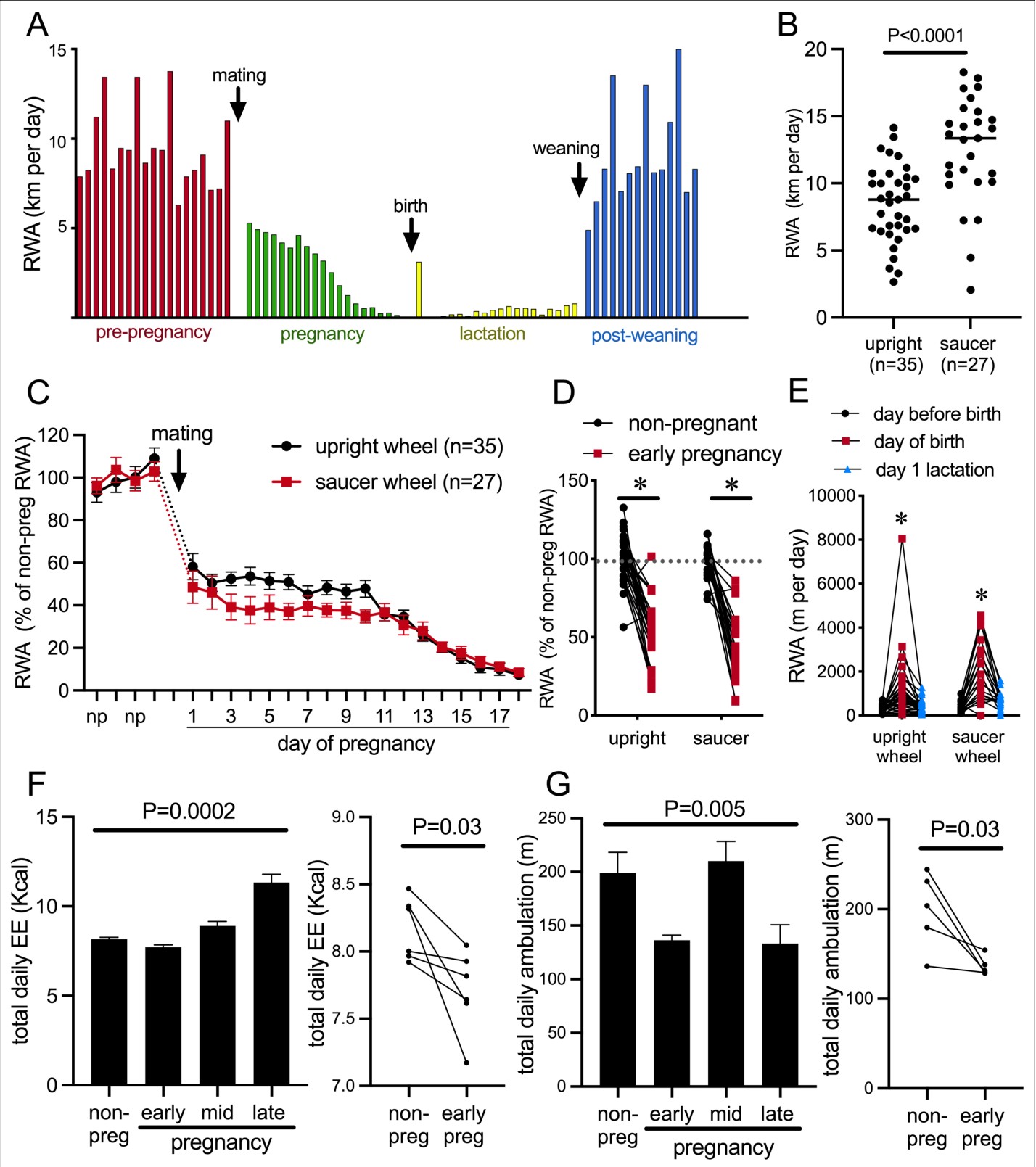

**Figure 1.** Pregnancy rapidly decreases voluntary running wheel activity (RWA). (**A**) Daily RWA of one representative mouse through one cycle of reproduction (pre-pregnancy estrous cycles, pregnancy, lactation, and after the weaning of her pups). Each bar represents RWA on 1 day. (**B**) RWA in non-pregnant, non-lactating (control state, termed 'non-pregnant') female mice with access to a traditional upright wheel or a low-profile saucer wheel. (**C**) RWA in female mice, with either traditional upright wheel or a low-profile saucer wheel, before and after successful mating. RWA activity for each

*Figure 1 continued on next page*

*Figure 1 continued*

mouse is expressed as a percentage of their average daily levels pre-pregnancy. (**D**) Daily RWA for the first three days of pregnancy expressed as a percentage of each mouse's RWA level pre-pregnancy. Dotted line indicates mean RWA level of non-pregnant groups, and * indicates a significant difference between non-pregnant RWA and early pregnancy RWA (effect of physiological state p<0.0001, *post hoc* non-pregnant vs. early pregnancy upright wheel: p<0.0001, saucer wheel: p<0.0001), n values same as C. (**E**) RWA for the 24 hr before giving birth, the 24 hr period that included birth, and the first day of lactation for mice housed with either an upright or saucer running wheel (effect of time p<0.0001, *post hoc* test: * indicates significantly different to other time points) (**F, G**) (*Left*) Bars represent total daily energy expenditure (EE; **F**) and total daily home cage ambulation (**G**) in the non-pregnant state and different time points during pregnancy (early = days 2–7, mid = days 8–13, late = days 14–18; repeated measures one-way ANOVA, n = 6). (*Right*) Points and lines represent the change in total daily EE (**F**) and total daily home cage ambulation (**G**) for each individual mouse between the non-pregnant state and early pregnancy (days 2–7; t-test, **F**: n = 6, **G**: n = 5).

lactation (*Figure 1A*), conditions characterized by high prolactin (*Phillipps et al., 2020*). To determine if prolactin can acutely influence physical activity, we investigated the effects of acute prolactin treatment on physical activity in non-pregnant mice. In addition to RWA in their home cages, a number of other indices of physical activity were assessed in various behavioral tests including both novel and familiar environments. C57BL/6J female mice (metestrous phase of the estrous cycle) that had been housed with a running wheel for at least 3 weeks were injected intraperitoneally (i.p.) with either prolactin (5 mg/kg) or vehicle 30 min before the start of the dark phase of the light/dark cycle and RWA was monitored. On the following metestrus, mice were injected with the alternative treatment (prolactin or vehicle) such that all mice received both treatment and control in a random order. Prolactin treatment led to a significant reduction in RWA (*Figure 2A*, repeated measures two-way ANOVA, interaction time × treatment p<0.0001, $F_{(13, 286)}$ = 6.684, and *Figure 2B*, paired t-test p = 0.007, t = 3.28, df = 11), particularly in the first 3 hr of the dark phase (*Figure 2A*, Sidak's multiple comparisons test: vehicle vs. prolactin 1 hr p=0.0009 [t = 5.107, df = 18.65], 2 hr p = 0.0173 [t = 3.736, df = 20.61], 3 hr p=0.0248 [t = 3.742, df = 15.91]) when female mice normally engage in their maximal levels of RWA (*De Bono et al., 2006*; *Manzanares et al., 2018*). No effect of acute prolactin treatment on RWA activity was observed in male mice (repeated measures two-way ANOVA, interaction time × treatment p=0.9222, $F_{(13, 286)}$ = 0.5028, and paired Student's t-test p=0.8732, t = 0.1634,

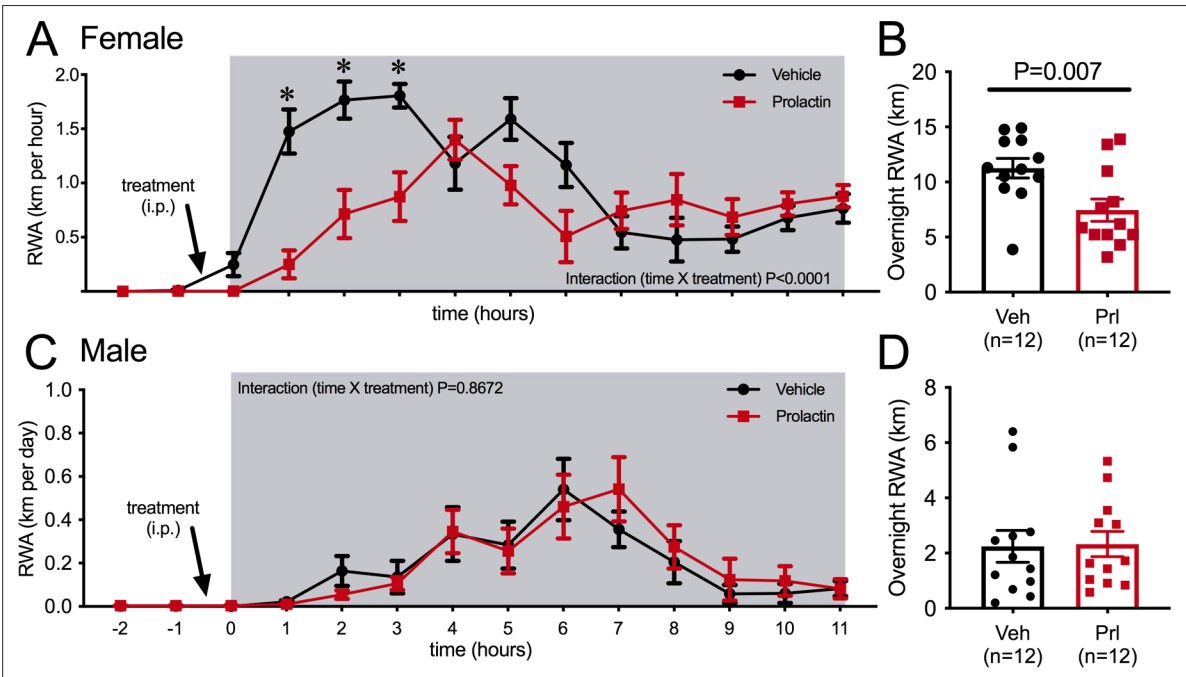

**Figure 2.** Acute effects of exogenous prolactin on physical activity. (**A**) Acute prolactin treatment significantly reduced running wheel activity (RWA) in female mice (**A, B**) but not in male mice (**C, D**). (**A, C**) Lines indicate RWA per hour in non-pregnant female mice (metestrous phase of estrous cycle for female mice) or male mice treated with either vehicle or prolactin (5 mg/kg intraperitoneally [i.p.]) 30 min before the start of the dark phase (dark phase indicated by gray shading). All mice (n = 12 per sex) received both treatments in a randomized order. * indicates time points that showed a significant difference (*post hoc* analysis p<0.05). (**B, D**) Bars represent total 12 hr RWA following either vehicle or prolactin (5 mg/kg i.p.) as described above.

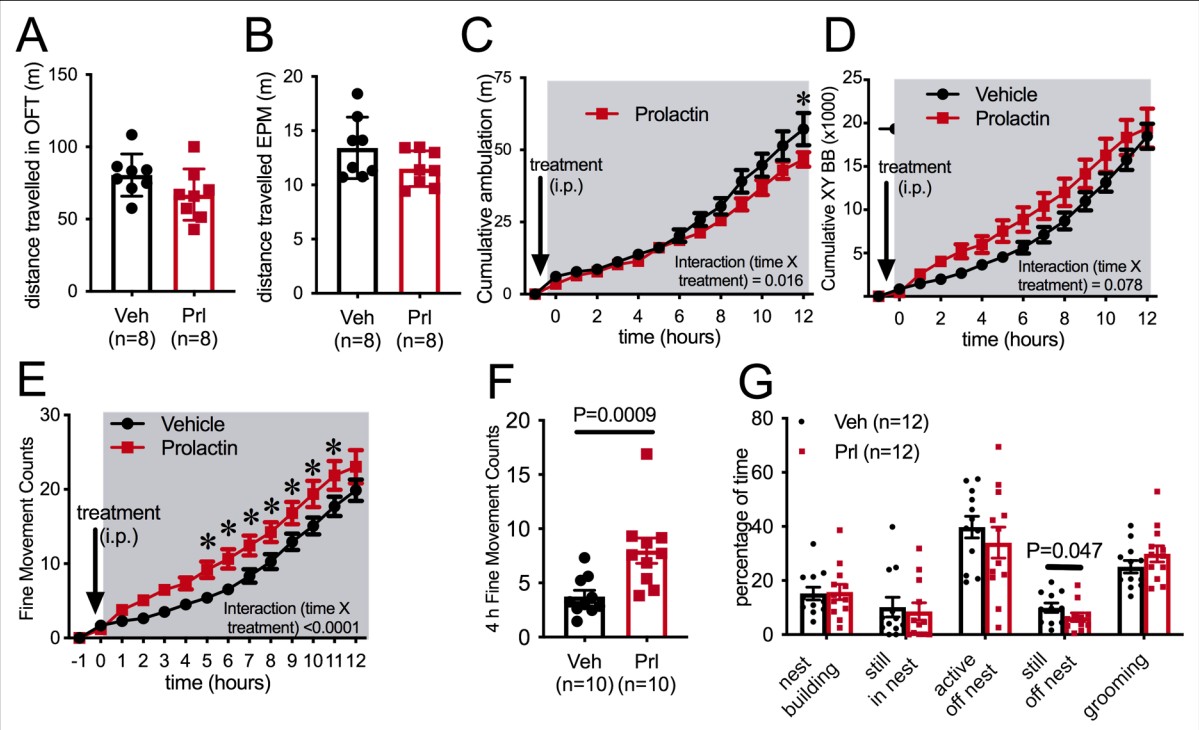

**Figure 3.** Prolactin does not acutely decrease ambulation. Bars represent total distanced traveled in an open field test (OFT) of 30 min duration (**A**) or elevated plus maze (EPM) of 5 min duration (**B**) carried out in the dark phase of the light cycle under dark conditions following either vehicle or prolactin (5 mg/kg intraperitoneally [i.p.]) treatment in female mice in the metestrous stage of the estrous cycle (n = 8 per group; t-test). (**C–F**) Acute prolactin had a subtle effect on ambulation in home cage. Lines represent cumulative ambulation (**C**), total X + Y beam breaks (**D**) and fine movement (activity that is not ambulation) (**E**) of female mice in the metestrous phase of the estrous cycle in home cage during the dark phase of the light cycle after vehicle or prolactin (5 mg/kg i.p.) treatment (repeated measures ANOVA). (**F**) Bars represent total fine movement counts in the first 4 hr following the start of the dark phase (paired t-test). (**G**) Acute prolactin did not greatly impact on time mice spent engaging in different behaviors in their home cages (paired t-test: 'still, off nest' vehicle vs. prolactin treatment p = 0.047 [n = 11 due to exclusion of data due to outlier; ROUT outlier test, outlier much higher value than the rest of the group], all other behaviors p>0.05). Bars represent percentage of time mice were engaged in various activities in their home cages (total test time 40 min) following vehicle or prolactin (5 mg/kg i.p.) treatment (n = 12 mice, all mice received both vehicle and prolactin in a randomized order, paired t-test).

df = 11; *Figure 2C and D*). In accordance with previously published work (reviewed in *Manzanares et al., 2018*), our C57BL/6J female mice ran significantly more per 24 hr than male mice (comparison between vehicle-treated females and males from *Figure 2B and D*, Student's t-test p<0.0001, t = 8.547, df = 22). The reason for this sex difference is unknown and requires further investigation.

In a novel environment, such as in an open field or the elevated plus maze (EPM) paradigm, prolactin injection an hour prior to testing did not influence distance traveled by female mice (*Figure 3A and B*), suggesting that there is not a generalized effect on locomotion (Student's t-test, EPM: p = 0.119, t = 1.661, df = 14, open field test [OFT]: p=0.1177, t = 1.667, df = 14). To enable more detailed investigation of physical activity levels within the familiar home cage environment, ambulatory movement was examined in C57BL/6J female mice housed in Promethion metabolic phenotyping cages, *without access to running wheels*. Mice were habituated to these cages, including handling and the associated removal from cages, for at least a week. On the metestrous phase of the estrous cycle, mice received an i.p. injection of either prolactin (5 mg/kg) or vehicle 30 min before the start of the dark phase of the light cycle, in a counterbalanced manner as described above. Prolactin treatment led to a significant reduction in ambulation during the 12 hr of the dark phase of the light cycle (*Figure 3C*, repeated measures ANOVA, interaction time × treatment, p=0.016, F(13, 234) = 2.079), although this was a subtle effect and only evident in the late stages of the dark phase (*Figure 3C*, Sidak's multiple comparisons test: vehicle vs. prolactin 12 hr p=0.03, t = 3.7, df = 12.6). The measure of 'ambulation distance' in these cages is generated by an algorithm that interprets multiple infrared beam breaks in a consecutive direction as forward movement or 'ambulation.' Interestingly, when total counts of XY beam

breaks were examined, prolactin treatment tended to increase, rather than decrease, total beam breaks, although there was no significant difference detected (*Figure 3D*, repeated measures ANOVA, interaction time × treatment p=0.078, F(13, 234) = 1.628). Further examination indicated that prolactin treatment significantly increased fine movement beam breaks, defined as beam breaks that are not in a consecutive direction, thus indicating non-ambulatory (stationary) movement (*Figure 3E*, repeated measures ANOVA, interaction time × treatment, p<0.0001, F(13, 117) = 183.9, *post hoc* analysis: vehicle vs. prolactin, p<0.05 for hours 5–12, t ranges from t = 3.038–4.119, df = 117). This increase in fine movement counts was observed both in the first 4 hr (*Figure 3F*, paired t-test p=0.0009, t = 4.882, df = 9) and the total 12 hr (paired t-test p=0.028, t = 2.619, df = 9, data not shown) of the dark phase. To determine what fine movement activities might be influenced by prolactin, another cohort of C57BL/6J female mice were video recorded in their home cages for 40 min following prolactin or vehicle injection (as described above). Recordings were analyzed to determine the time spent in various activities, such as nest building and grooming. There was no difference identified in the time spent doing any of these behaviors following prolactin or vehicle injection except for time spent 'still, off nest,' which was slightly yet significantly reduced in the prolactin-treated group (*Figure 3G*, paired Student's t-tests, 'still, off nest': p = 0.047, t = 2.261, df = 10). Overall, this set of experiments demonstrated that prolactin reduces voluntary RWA in female mice, but had little or no significant impact on general ambulation levels in either a novel environment or in the home cage. Therefore, while rapid changes in RWA activity in early pregnancy may be driven by mating-induced prolactin surges, there is no evidence to support a similar mechanism underlying reductions in ambulation in early pregnancy (*Figure 1F*).

## Prolactin receptors in forebrain neurons are necessary for pregnancy-induced suppression of RWA

To test the hypothesis that prolactin is acting in the brain to mediate the pregnancy-induced suppression of RWA, we measured RWA in a mouse line lacking Prlr in most forebrain neurons (*Prlr*^lox/lox^ mice crossed with CamKinase2a-Cre mice [*Prlr*^lox/lox^/*Camk2a*^Cre^], as previously described *Brown et al., 2016*; see *Figure 4—figure supplement 1A* for validation of Prlr deletion in these mice). In mice, Prlr are highly expressed in the hypothalamus, mostly localized to the arcuate, ventromedial (VMN), paraventricular (PVN), anteroventral periventricular nuclei, and the preoptic area (POA) (*Brown et al., 2010*; *Kokay et al., 2018*). Within the POA, there is high expression of Prlr in the median, periventricular, and medial regions. Lower expression is observed in the anterior hypothalamic area and dorsomedial hypothalamus. Outside of the hypothalamus, the medial amygdala, anterior bed nucleus of stria terminalis (BNST), lateral septum, and periaqueductal gray also have robust Prlr expression (*Brown et al., 2010*). Expression of Prlr in the hypothalamus does not appear to undergo significant modification during pregnancy and lactation, except for an increase in the arcuate nucleus detected in late pregnancy in the rat (*Augustine et al., 2003*). During pregnancy, elevated levels of phosphorylation of the transcription factor signal transducer and activator of transcription 5 (pSTAT5) are detected throughout the brain. This is a well-characterized intracellular signaling molecule-activated downstream of the Prlr (and a range of other cytokines), but we have shown that the majority of pSTAT5 in the brain during pregnancy is caused by increased Prlr activation (*Gustafson et al., 2020*), both by surges in prolactin early in pregnancy and by rising placental lactogen as pregnancy progresses (*Phillipps et al., 2020*). During lactation, various regions within the brain show increased prolactin-induced pSTAT5 expression compared to mice in the diestrous phase of the estrous cycle, including the medial POA, BNST, PVN, and medial amygdala (*Brown et al., 2011*).

Our *Prlr*^lox/lox^ mouse line is constructed such that in the presence of Cre-recombinase the sequence between the loxP sites of the *Prlr* gene undergoes inversion, resulting in the deletion of exons 5–10 of the *Prlr* gene and induced expression of enhanced green fluorescent protein (GFP) (*Brown et al., 2016*). Thus, in this mouse line, GFP is a useful marker of cells in which Cre-mediated recombination (and therefore *Prlr* deletion) has taken place. Similar to our previous work, here we show that while *Prlr*^lox/lox^/*Camk2a*^Cre^ do not have a complete Prlr deletion in the forebrain, there are areas of extensive deletion (as measured by reduced prolactin-induced pSTAT5), such as the arcuate nucleus and VMN, and areas where Prlr is reduced by about 50% such as the medial POA (MPOA) (*Figure 4—figure supplement 1A*, Mann–Whitney non-parametric test MPOA: p=0.0262, U = 7, Arc: p=0.0002, U = 0, VMN: p = 0.0071, U = 7; *Brown et al., 2016*; *Gustafson et al., 2020*). In the MPOA, there was

a significant reduction of Prlr from GABAergic neurons (*Figure 4—figure supplement 1B*, Mann–Whitney non-parametric test, p=0.0091, U = 0). As previously shown, Prlr is not significantly reduced in the PVN in the *Prlr*$^{lox/lox}$/*Camk2a*$^{Cre}$ mice (*Figure 4—figure supplement 1A*, Mann–Whitney non-parametric test PVN: p=0.6065, U = 26; *Gustafson et al., 2020*).

RWA was monitored for at least 2 weeks in the non-pregnant state, then mice were mated and RWA was monitored throughout pregnancy, along with body weight and food intake. *Prlr*$^{lox/lox}$/*Camk2a*$^{Cre}$ mice do not show normal 4- to 5 -day estrous cycles (*Figure 4A* and *Figure 4—figure supplement 3A*, Mann–Whitney test, p<0.0001, U = 0) due to hyperprolactinemia caused by a lack of negative feedback in the hypothalamus (*Brown et al., 2016*). The high prolactin acts in the ovary to prolong progesterone secretion after each ovulation, resulting in serial pseudopregnancy-like phases of around 10–12 days between ovulations (*Figure 4—figure supplement 3A*). We observed a trend for reduced RWA in non-pregnant *Prlr*$^{lox/lox}$/*Camk2a*$^{Cre}$ compared to control mice (*Prlr*$^{lox/lox}$) in the same state (*Figure 4B*, t-test, p=0.058, t = 1.984, df = 27), and in particular, the absence of the cyclical pattern of elevated RWA prior to ovulation that has previously been described (*Ladyman et al., 2018a*; *Basterfield et al., 2009*; *Lightfoot, 2008*; *Rosenfeld, 2017*; *Figure 4C*). The cumulative RWA activity over 5 days (to take into account the 4- to 5 -day estrous cycle observed in our control mice) showed no difference between genotypes (*Figure 4D*, p=0.0831, t = 1.802, df = 26), suggesting that the absence of estrous cycle-induced changes in the RWA of *Prlr*$^{lox/lox}$/*Camk2a*$^{Cre}$ did not significantly impact on total amount of RWA. Despite the abnormal cycles and slightly increased body weight (Student's t-test, p=0.0156, t = 2.570, df = 29) in the non-pregnant state (*Figure 4—figure supplement 3A and B*), these mice are able to become pregnant and maintain a pregnancy, showing no differences in fetus number (Mann–Whitney test, p=0.1911, U = 16.5) and uterus mass in late pregnancy (Student's t-test p=0.4416, t = 0.7938, df = 13) or litter size on day 4 of lactation (Mann–Whitney test, p=0.1568, U = 83.5; *Figure 4—figure supplement 3C–E*).

Mice lacking Prlr in most forebrain neurons (*Prlr*$^{lox/lox}$/*Camk2a*$^{Cre}$) failed to show the characteristic decrease in RWA in early pregnancy, and in fact showed significantly increased RWA compared to control mice throughout pregnancy (*Figure 4E*, interaction genotype × time, p<0.0001, F(21, 481) = 3.297). When analyzing the overall change in RWA across the first week of pregnancy, *Prlr*$^{lox/lox}$/*Camk2a*$^{Cre}$ mice did not show a significant reduction in RWA, whereas control mice reduced RWA to approximately 50% of non-pregnant levels in the first few days of pregnancy (*Figure 4F*, interaction genotype × time, p=0.0385, F(1,41) = 4.57, Sidak's multiple comparisons test between non-pregnant and early pregnant: *Prlr*$^{lox/lox}$ p=0.002 [t = 4.310, df = 41], *Prlr*$^{lox/lox}$/*Camk2a*$^{Cre}$ p=0.3899 [t = 1.249, df = 41]). All pregnant mice showed a reduction in RWA over the course of pregnancy, but even when heavily pregnant on days 15–18 of pregnancy, the *Prlr*$^{lox/lox}$/*Camk2a*$^{Cre}$ animals continued to run significantly more on the wheel than controls (*Figure 4G and H*, day 15 hourly RWA pattern: interaction time × genotype, p<0.0001, F(23, 621) = 4.363; total RWA on day 15, Student's t-test p=0.0035, t = 3.215, df = 26, n = 12–15 per genotype [data not shown]; days 16–18 cumulative RWA: Student's t-test p<0.0001, t = 5.037, df = 23). These data suggest that while weight gain and physical constraints are likely to play a major role in the overall suppression of RWA in mid-late pregnancy, prolactin action in the brain still plays a contributing role even at this stage of pregnancy. Analyzing late pregnancy (day 17) RWA with body weight as a covariate demonstrated a significant difference, independent of body weight (ANCOVA p=0.0006, F(1,24) = 15.58). The slightly lower gestational weight gain (*Figure 4—figure supplement 3F*, interaction genotype × time, p<0.0001, F(17,325) = 3.348) and slightly elevated pre-pregnancy body weight in the *Prlr*$^{lox/lox}$/*Camk2a*$^{Cre}$ (*Figure 4—figure supplement 3B*) meant that by late pregnancy body mass was not significantly different between *Prlr*$^{lox/lox}$/*Camk2a*$^{Cre}$ and controls (*Figure 4—figure supplement 3G*, Student's t-test p=0.7885, t = 0.2711, df = 25), meaning that differences in RWA in late pregnancy between genotypes cannot be attributed to body weight. In the presence of a running wheel, *Prlr*$^{lox/lox}$/*Camk2a*$^{Cre}$ mice gained slightly, yet significantly, less weight by the later stages of pregnancy (*Figure 4—figure supplement 3F*) compared to controls. Both *Prlr*$^{lox/lox}$/*Camk2a*$^{Cre}$ and control mice showed a pregnancy-induced increase in food intake (effect of pregnancy p<0.0001, F(17, 368) = 7.992), but there was no difference in food intake in *Prlr*$^{lox/lox}$/*Camk2a*$^{Cre}$ compared to control mice (*Figure 4—figure supplement 3H*, effect of genotype p=0.8177, F(1,368) = 0.0532).

To eliminate the possibility that the loss of Prlr in the brain might cause exaggerated stress responses in the pregnant females (*Slattery and Neumann, 2008*; *Torner et al., 2001*) that could

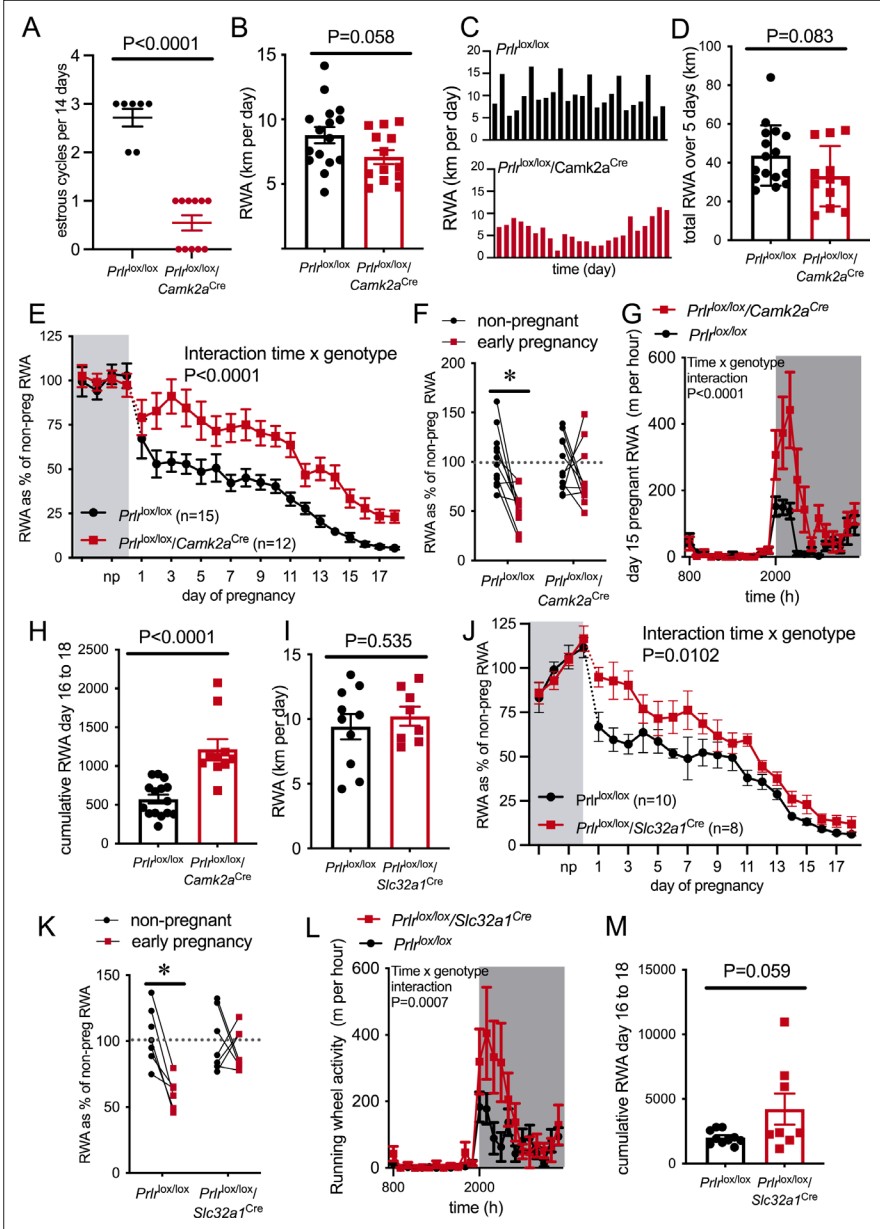

**Figure 4.** Pregnancy-induced suppression of running wheel activity (RWA) is attenuated in mice lacking Prlr in forebrain neurons. (**A**) Number of estrous cycles in a continuous 12 -day period in $Prlr^{lox/lox}/Camk2a^{Cre}$ mice (n = 7–11). (**B**) Average daily RWA in non-pregnant state ($Prlr^{lox/lox}/Camk2a^{Cre}$ n = 14, $Prlr^{lox/lox}$, n = 15). (**C**) Representative examples of daily RWA in one control ($Prlr^{lox/lox}$) and one $Prlr^{lox/lox}/Camk2a^{Cre}$ mouse over a 24 -day period. (**D**) Cumulative RWA over the 5 days prior to mating in $Prlr^{lox/lox}/Camk2a^{Cre}$ (n = 14) and controls ($Prlr^{lox/lox}$, n = 15). (**E**) RWA activity before and after successful mating in $Prlr^{lox/lox}/Camk2a^{Cre}$ mice. RWA activity for each mouse is expressed as a percentage of their individual average daily levels in the non-pregnant state ($Prlr^{lox/lox}/Camk2a^{Cre}$ n = 12, $Prlr^{lox/lox}$, n = 15, total daily RWA per animal shown in ***Figure 4—source data 1***). (**F**) Change in RWA from non-pregnant state to early pregnancy in $Prlr^{lox/lox}/Camk2a^{Cre}$ mice. Dotted line indicates mean non-pregnant RWA for each group. (**G**) 24 hr RWA profile from day 15 pregnant $Prlr^{lox/lox}/Camk2a^{Cre}$ mice (n = 13) and controls (n = 15), RWA per animal shown in ***Figure 4—source data 3***. Gray indicates dark phase of light cycle. (**H**) Cumulative RWA over days 16–18 pregnancy ($Prlr^{lox/lox}/Camk2a^{Cre}$ n = 10, $Prlr^{lox/lox}$, n = 15). (**I**) Average daily RWA in non-pregnant state ($Prlr^{lox/lox}/Slc32a1^{Cre}$ n = 8, $Prlr^{lox/lox}$, n = 10). (**J**) RWA activity before and after successful mating in $Prlr^{lox/lox}/Slc32a1^{Cre}$ mice. RWA activity for each mouse is expressed as a percentage of their individual average daily levels in the non-pregnant state ($Prlr^{lox/lox}/Slc32a1^{Cre}$, n = 8, $Prlr^{lox/lox}$, n = 10, total daily RWA per animal shown in ***Figure 4—source data 2***). (**K**) Change in RWA from non-pregnant state to early pregnancy in $Prlr^{lox/lox}/Slc32a1^{Cre}$ mice. Dotted line indicates mean non-pregnant RWA for each group. (**L**) 24 hr RWA profile from day 15 pregnant

*Figure 4 continued on next page*

*Figure 4 continued*

$Prlr^{lox/lox}/Slc32a1^{Cre}$ mice (n = 8) and controls (n = 10), RWA per animal shown in ***Figure 4—source data 3***. Gray indicates dark phase of light cycle. (**M**) Cumulative RWA over days 16–18 pregnancy ($Prlr^{lox/lox}/Slc32a1^{Cre}$, n = 8, $Prlr^{lox/lox}$, n = 10).

The online version of this article includes the following figure supplement(s) for figure 4:

**Source data 1.** Each line (gray = control, red = $Prlr^{lox/lox}/Camk2a^{Cre}$) represents total daily running wheel activity (RWA) from each individual mouse in the non-pregnant state (4 days before mating) and during pregnancy (days 1–18).

**Source data 2.** Each line (gray = control, red = $Prlr^{lox/lox}/Slc32a1^{Cre}$) represents total daily running wheel activity (RWA) from each individual mouse in the non-pregnant state (4 days before mating) and during pregnancy (days 1–18).

**Source data 3.** Each line black = control, red = $Prlr^{lox/lox}/Camk2a^{Cre}$ (top) or $Prlr^{lox/lox}/Slc32a1^{Cre}$ (bottom) represents hourly running wheel activity (RWA) from each individual mouse on day 15 of pregnancy.

**Figure supplement 1.** Conditional deletion of Prlr in forebrain neurons.

**Figure supplement 2.** Conditional deletion of Prlr in *Slc32a1*-expressing neurons.

**Figure supplement 3.** Pregnancy parameters of mice with conditional deletion of Prlr in most forebrain neurons or in GABA neurons.

influence acute RWA (***Malisch et al., 2016***), we also examined anxiety-like behavior in late pregnant (day 15/16) $Prlr^{lox/lox}/Camk2a^{Cre}$ and control mice using the EPM, as described previously (***Ladyman et al., 2018b***). Time spent in the open arms of the EPM was similar in both groups, suggesting no difference in anxiety-like behavior in the pregnant $Prlr^{lox/lox}/Camk2a^{Cre}$ and control mice (***Figure 4—figure supplement 3I***, Student's t-test p=0.7756, t = 0.291, df = 3).

While the data showing a lack of rapid pregnancy-induced suppression of RWA in $Prlr^{lox/lox}/Camk2a^{Cre}$ were clear, the elevated prolactin (***Brown et al., 2016***) and abnormal estrous cycles (***Figure 4A***) were a potential confound in this mouse line. Hence, we aimed to replicate these experiments using a mouse line with a specific deletion of Prlr from GABA neurons ($Prlr^{lox/lox}$ crossed with mice expressing Cre recombinase in GABA neurons [$Slc32a1^{Cre}$], as previously described ***Brown et al., 2016***; ***Figure 4—figure supplement 2***). These mice have more limited, although still extensive, deletion of prolactin receptors throughout the brain (***Figure 4—figure supplement 2***, Mann–Whitney non-parametric test MPOA: p=0.0464, U = 15, Arc: p = 0.0014, U = 6; ***Brown et al., 2016***; ***Brown et al., 2017***). In the MPOA, we have previously shown that approximately half the prolactin-responsive neurons are GABAergic (***Brown et al., 2017***). In the present study, we found that $Prlr^{lox/lox}/Slc32a1^{Cre}$ mice had a significant reduction in the number of GABA neurons that co-localize with Prlr (around 70%, Mann–Whitney non-parametric test p=0.0022, U = 0); however, it should be noted that this was not a complete deletion of Prlr from all GABAergic neurons. Importantly, however, Prlr are not deleted from the majority of hypothalamic dopamine neurons in this line, meaning that prolactin levels are normal and they have normal 4- to 5 -day estrous cycles (***Brown et al., 2016***). $Prlr^{lox/lox}/Slc32a1^{Cre}$ mice have similar body weight in the virgin state as controls (***Figure 4—figure supplement 3J***, Student's t-test p=0.3962, t = 0.8718, df = 16), can carry a pregnancy to term, and both genotypes had similar sized litters (***Figure 4—figure supplement 2K***, Mann–Whitney p = 0.7734, U = 41).

Non-pregnant $Prlr^{lox/lox}/Slc32a1^{Cre}$ and control ($Prlr^{lox/lox}$) mice had similar levels of RWA (***Figure 4I***, Student's t-test p=0.5358, t = 0.6329, df = 16). In contrast, however, $Prlr^{lox/lox}/Slc32a1^{Cre}$ mice had markedly higher RWA throughout pregnancy compared to controls (***Figure 4F***, interaction time × genotype, p=0.010, F(21, 291) = 1914). Consistent with observations in $Prlr^{lox/lox}/Camk2a^{Cre}$ mice, $Prlr^{lox/lox}/Slc32a1^{Cre}$ mice did not show the rapid reduction in RWA in early pregnancy, whereas control mice showed the characteristic pregnancy-induced immediate reduction in RWA (***Figure 4G***, interaction genotype × time, p=0.0162, F(1, 23) = 6.735, Sidak's multiple comparisons test for significant effect of time: $Prlr^{lox/lox}$, p=0.0007, t = 4.196, df = 23, $Prlr^{lox/lox}/Slc32a1^{Cre}$: p = 0.7888, t = 0.6215, df = 23). Again, similar to $Prlr^{lox/lox}/Camk2a^{Cre}$, no differences in anxiety-like behavior were detected in pregnant $Prlr^{lox/lox}/Slc32a1^{Cre}$ mice (***Figure 4—figure supplement 3L***, Student's t-test p=0.8617, t = 0.1762, df = 22). While food intake and body weight increased across pregnancy (***Figure 4—figure supplement 3M***: body weight, effect of time, p<0.0001, F(1.413, 11.89) = 302.9; ***Figure 4—figure supplement 3N***: food intake, effect of time, p<0.0001, F(5.283, 60.6) = 10.09), there was no difference

in body weight gain or food intake during pregnancy in $Prlr^{lox/lox}/Slc32a1^{Cre}$ mice compared to controls (**Figure 4—figure supplement 3**, body weight gain: effect of genotype, p = 0.4648, F(1, 15) = 0.5627; food intake: effect of genotype, p=0.3772, F(1,17) = 0.8224). Overall, the attenuated reduction in RWA in early pregnancy in both $Prlr^{lox/lox}/Camk2a^{Cre}$ and $Prlr^{lox/lox}/Slc32a1^{Cre}$ strongly indicates a role for prolactin action in the brain, driving early changes in RWA as soon as pregnancy is initiated.

On day 15 of pregnancy, $Prlr^{lox/lox}/Slc32a1^{Cre}$ mice engaged in more RWA than controls (**Figure 4L**, day 15 hourly RWA pattern, interaction time × genotype, p=0.0007, F(23, 391) = 2.291; total RWA on day 15, Student's t-test p=0.0494, t = 2.116, df = 17, n = 9–10; data not shown). Unlike the continued higher RWA seen in the $Prlr^{lox/lox}/Camk2a^{Cre}$ mice, as pregnancy advanced further, RWA became similar between $Prlr^{lox/lox}/Slc32a1^{Cre}$ mice and controls, as shown by no significant difference in cumulative RWA over days 16–18 of pregnancy (days 16–18 cumulative RWA: Mann–Whitney non-parametric test p=0.1728, U = 24). Body weight in late pregnancy was not different between $Prlr^{lox/lox}/Slc32a1^{Cre}$ mice and controls ( **Figure 4—figure supplement 3O**, day 17 pregnant body weight Student's t-test p=0.1742, t = 1.437, df = 13). These results suggest that in late pregnancy prolactin-induced suppression of RWA is not mediated by GABAergic neurons.

## Identifying the neuronal populations involved in prolactin-induced suppression of RWA during pregnancy

Based on a report that ablation of arcuate nucleus kisspeptin neurons markedly suppressed RWA in female mice (**Padilla et al., 2019**), we next examined the role of these neurons in mediating the pregnancy-induced decrease in RWA. We have shown that arcuate kisspeptin neurons express Prlr (**Araujo-Lopes et al., 2014**; **Kokay et al., 2011**) and that prolactin is likely to inhibit their activity (**Brown et al., 2019**), although it should be noted that only a small proportion of these kisspeptin cells are GABAergic (**Marshall et al., 2017**). Mice with Prlr deleted from kisspeptin neurons were generated by crossing $Kiss1^{Cre}$ mice (**Mayer et al., 2010**) with $Prlr^{lox/lox}$ mice ($Prlr^{lox/lox}/Kiss1^{Cre}$) (**Brown et al., 2019**). We have previously shown that these mice have normal estrous cycles and have Prlr deleted specifically in the arcuate nucleus kisspeptin neurons as demonstrated by in situ hybridization for *Prlr* and *Kiss1* mRNAs and also by the presence of enhanced GFP using immunohistochemistry (**Figure 5A**, Mann–Whitney non-parametric test p=0.0004, U = 0; **Brown et al., 2019**).

In the non-pregnant state, $Prlr^{lox/lox}/Kiss1^{Cre}$ and control $Prlr^{lox/lox}$ mice did not differ in their mean daily RWA (**Figure 5B**, p=0.6604, t = 0.4472, df = 17). Overall, there was a significant effect of pregnancy on RWA activity in both $Prlr^{lox/lox}/Kiss1^{Cre}$ and control $Prlr^{lox/lox}$ mice (**Figure 5C**, effect of time, p<0.0001, F(4.649, 60.88) = 39.98) but no difference between the genotypes (effect of genotype, p=0.8805, F(1, 14) = 0.02345). In other words, both genotypes showed the normal pregnancy-induced suppression in RWA in early pregnancy, and there was no major effect of deletion of Prlr in arcuate kisspeptin neurons. On closer analysis of the early pregnancy change in RWA, while both genotypes showed a decrease in RWA behavior early in pregnancy (**Figure 5D**, interaction genotype × early pregnancy, p=0.01, F(1, 18) = 7.559, Sidak's multiple comparisons test for significant effect of time: $Prlr^{lox/lox}$, p<0.0001, t = 9.307, df = 28; $Prlr^{lox/lox}/Kiss1^{Cre}$, p<0.0001, t = 5.419, df = 28), the effect of pregnancy on RWA in the first few days of pregnancy (days 1–3) was subtly reduced in $Prlr^{lox/lox}/Kiss1^{Cre}$ mice compared with control mice (**Figure 5D**, interaction genotype × early pregnancy, p=0.01, F(1, 18) = 7.559, Sidak's multiple comparisons test for significant effect of genotype: early pregnancy, p=0.0049, t = 3.329, df = 28). Thus, these data suggest the possibility that prolactin action on arcuate kisspeptin neurons may play a small contributing role in the initial reduction in RWA during pregnancy, but this is not the major factor in mediating this response. Later in pregnancy, $Prlr^{lox/lox}/Kiss1^{Cre}$ mice and controls did not differ in their RWA (**Figure 5B**, total RWA day 15: p = 0.8949, t = 0.1345, df = 14; cumulative RWA days 16–18: p=0.7656, t = 0.3041, df = 14).

Finally, we sought to narrow down the site of action for prolactin in mediating the effect on RWA during early pregnancy. Our previous work has identified the medial POA as a key site mediating prolactin action on behavioral adaptations associated with pregnancy (**Brown et al., 2017**). This region has also been linked to the hormonal regulation of RWA during the estrous cycle (**Grigsby et al., 2020**; **King, 1979**). The MPOA contains the largest concentration of Prlr-expressing neurons in the hypothalamus (**Brown et al., 2010**), and over 50% of these neurons are GABAergic (**Brown et al., 2017**). Moreover, RWA is inherently rewarding for rodents (**Novak et al., 2012**), and we have previously identified a Prlr-sensitive projection from the MPOA to the VTA (**Brown et al., 2017**), potentially

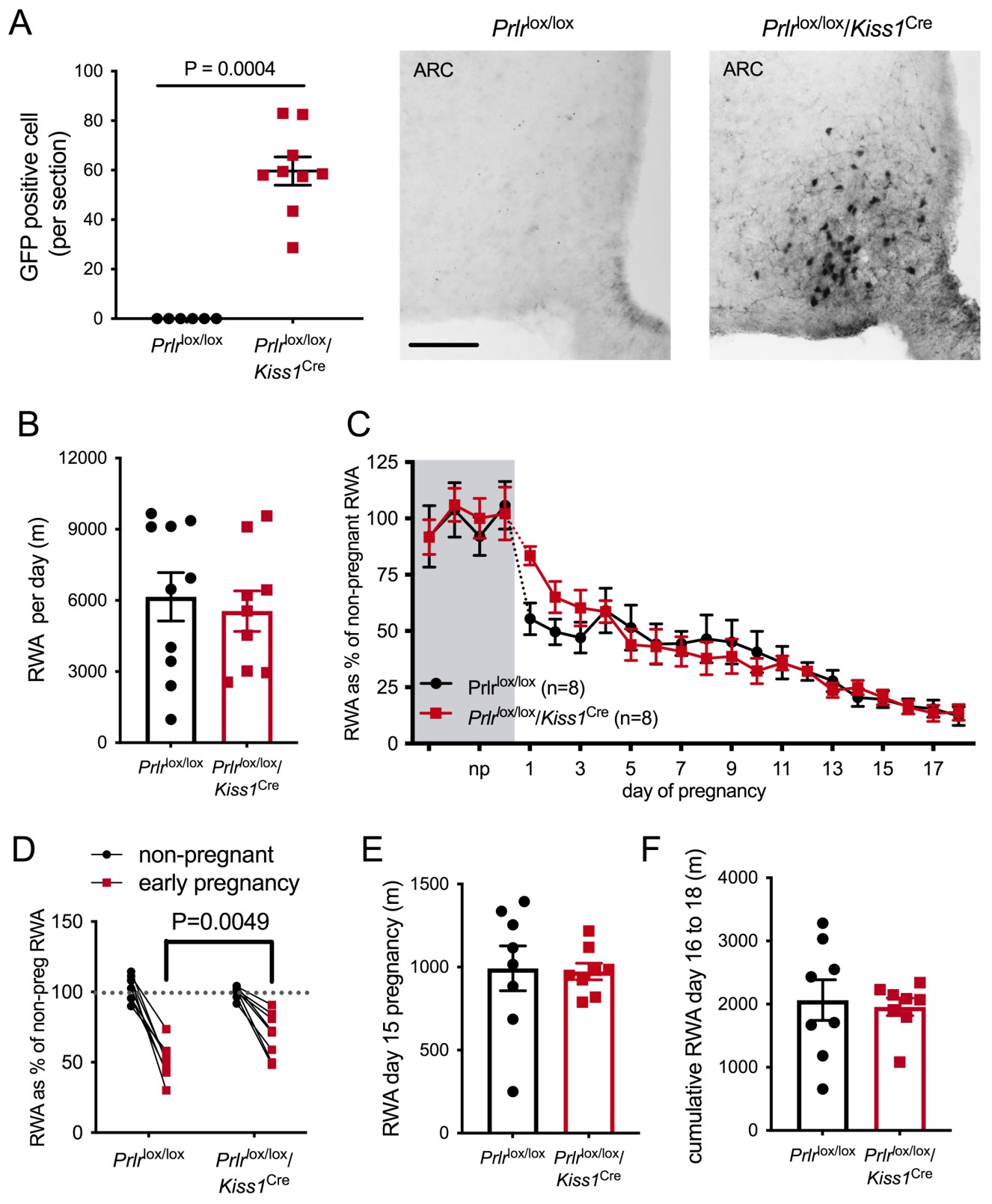

**Figure 5.** Effect of Prlr deletion from arcuate nucleus kisspeptin neurons on running wheel activity (RWA) during pregnancy. (**A**) Data represent number of green fluorescent protein (GFP)-positive cells (black staining), indicative of Prlr gene deletion, in the arcuate nucleus of *Prlr*lox/lox/*Kiss1*Cre female mice alongside representative images, n = 6–9 per group (scale bar = 50 µm). (**B**) Average daily RWA in non-pregnant state (*Prlr*lox/lox/*Kiss1*Cre, n = 9, *Prlr*lox/lox, n = 10). (**C**) RWA activity before and after successful mating in *Prlr*lox/lox/*Kiss1*Cre mice. RWA activity for each mouse is expressed as a percentage of their

Figure 5 continued

individual average daily levels in the non-pregnant state ($Prlr^{lox/lox}/Kiss1^{Cre}$ n = 8, $Prlr^{lox/lox}$ n = 8, total daily RWA per animal shown in **Figure 5—source data 1**). (**D**) Change in RWA from non-pregnant state to early pregnancy in $Prlr^{lox/lox}/Kiss1^{Cre}$ mice. Dotted line indicates mean non-pregnant RWA for each group. (**E**) RWA on day 15 pregnancy for $Prlr^{lox/lox}/Kiss1^{Cre}$ mice (n = 8) and controls (n = 8). (**F**) Cumulative RWA over days 16–18 pregnancy ($Prlr^{lox/lox}/Kiss1^{Cre}$, n = 8, $Prlr^{lox/lox}$, n = 8). Note that in contrast to $Prlr^{lox/lox}/Camk2a^{Cre}$ and $Prlr^{lox/lox}/Slc32a1^{Cre}$ models, above, there was no sustained difference in levels of RWA in mid and late pregnancy.

The online version of this article includes the following figure supplement(s) for figure 5:

**Source data 1.** Each line (gray = control, red = $Prlr^{lox/lox}/Kiss1^{Cre}$) represents total daily running wheel activity (RWA) from each individual mouse in the non-pregnant state (4 days before mating) and during pregnancy (days 1–18).

impacting on dopamine-mediate reward pathways. Hence, we hypothesized that prolactin action on these behavioral circuits originating in the MPOA may mediate the suppressive effect on RWA.

To investigate this possibility, we used an adeno-associated virus (AAV) to deliver Cre recombinase specifically into the MPOA of adult female $Prlr^{lox/lox}$ mice, with controls consisting of AAV-mCherry delivered to $Prlr^{lox/lox}$ mice. Following recovery from stereotaxic surgery for virus delivery, mice were housed with a running wheel for at least 3 weeks, then mated, and RWA was monitored throughout pregnancy. Similar to what we have reported previously, AAV-cre injection into the MPOA of $Prlr^{lox/lox}$ mice resulted in Cre-mediated recombination expression of GFP (**Figure 6A**) and removed all functional prolactin responses from this region, as determined by prolactin-induced pSTAT5 (**Figure 6A**; **Brown et al., 2017**). Replicating work we have published previously, this manipulation also led to the mothers abandoning their pups following birth (**Figure 6B**, number of live pups on day 3 lactation: Student's t-test p<0.0001, t = 10, df = 15), confirming the critical role of Prlr in this area for normal maternal behavior in lactation (**Brown et al., 2017**). Mice with a specific deletion of Prlr in the MPOA did not show the immediate change in RWA once pregnant (**Figure 6C**, interaction virus injection × day of pregnancy, p<0.0001, F(21, 303) = 3.451). Strikingly, mice lacking Prlr in the MPOA did not demonstrate any reduction in RWA in the first 3 days of pregnancy a time when dramatic reductions are occurring in controls (**Figure 6C**, interaction virus injection × early pregnancy, p=0.0267, F(1, 29) = 5.447, Sidak's multiple comparisons test for effect of early pregnancy: control injection p=0.01, t = 3.035, df = 29; AAV-cre injection, p=0.9687, t = 0.2256, df = 29). As pregnancy progressed, RWA remained higher in mice with a specific deletion of Prlr in the MPOA compared to controls (**Figure 6F**: day 15 pregnant RWA: Student's t-test p=0.0036, t = 3.442, df = 15; **Figure 6G**: days 16–18 pregnancy cumulative RWA: Mann–Whitney non-parametric test, p = 0.0079, U = 9). These data indicate that the MPOA is a key site for prolactin-induced suppression of RWA during early pregnancy and for the maintained prolactin-induced contribution to attenuation of RWA throughout pregnancy.

While the main focus of this work was on investigating if prolactin contributes to the rapid suppression of RWA immediately following mating, it was also of interest to note that RWA also remains extremely low throughout lactation (**Figure 1A**; **Ladyman et al., 2018a**) when prolactin is high (**Phillipps et al., 2020**). To assess if prolactin also contributes to the suppression of RWA during lactation, RWA activity was monitored across lactation in one cohort of $Prlr^{lox/lox}/Camk2a^{Cre}$ mice (n = 5) and control mice ($Prlr^{lox/lox}$) (n = 6). Where we had collected data, RWA was also evaluated for the first 3 days of lactation in a larger cohort of $Prlr^{lox/lox}/Camk2a^{Cre}$ mice and $Prlr^{lox/lox}/Slc32a1^{Cre}$ mice (and their respective littermate controls). As reported previously, RWA was low throughout lactation, apart from a small rise towards the last few days before weaning (**Figure 7A**). It should be noted that in the last week of lactation it is likely that offspring may interact with the running wheel and it is impossible to determine in these data if any of the RWA is due to offspring and not the mother.

$Prlr^{lox/lox}/Camk2a^{Cre}$ mice had significantly higher RWA in early lactation compared to controls (**Figure 7A and C**, effect of genotype: p<0.0001, F(1, 22) = 29.08, Sidak's multiple comparisons test for difference between genotype: day 1, p=0.0025, t = 4.472, df = 11.51; day 2, p=0.0057, t = 4.223, df = 9.628; day 3, p=0.0258, t = 3.391, df = 8.506) despite similar pup numbers (**Figure 7B**, Mann–Whitney non-parametric test p=0.2078, U = 8). For most of lactation, however, there was no difference in RWA between the knockout animals and the controls (**Figure 7A**, interaction time × genotype: p=0.0194, Sidak's multiple comparisons test for difference between genotype: day 1, p=0.0196, t = 3.434, df = 72; days 2–20, p>0.05), suggesting that prolactin action is not necessary for the sustained suppression of RWA in lactation. Interestingly, three out of five $Prlr^{lox/lox}/Camk2a^{Cre}$ mice each showed extremely high RWA on 1 day in mid lactation (days 6, 8, and 11, respectively), indicating that without

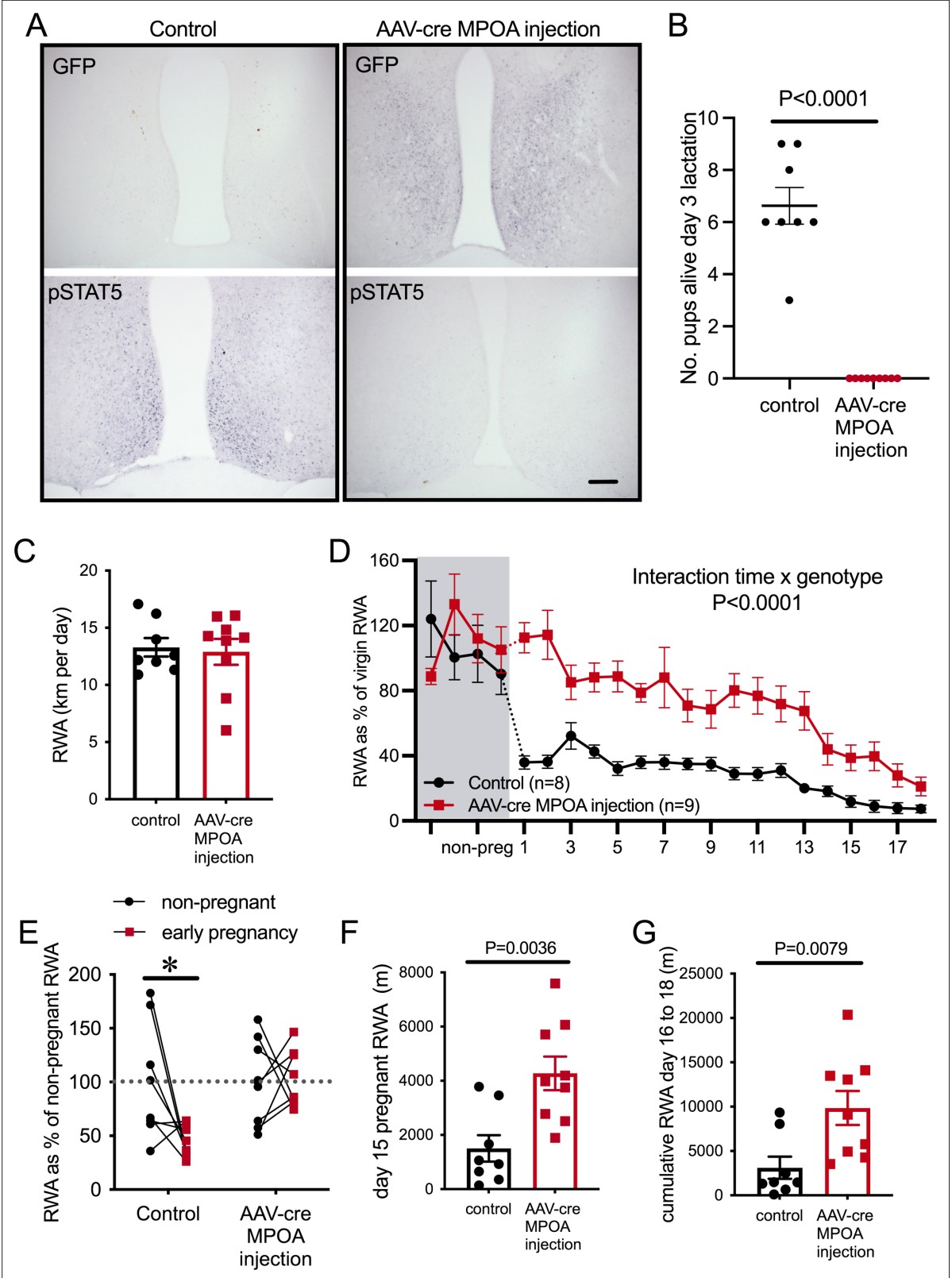

**Figure 6.** Prolactin-sensitive neurons in the medial preoptic area (MPOA) mediate pregnancy-induced suppression of running wheel activity (RWA). (**A**) Representative images of coronal sections through the MPOA showing immunohistochemical staining for green fluorescence protein (GFP) (*top*) and prolactin-induced pSTAT5 (*bottom*). Control injection *Prlr*<sup>lox/lox</sup> mice showed an absence of GFP and extensive pSTAT5 while AAV-cre-injected *Prlr*<sup>lox/lox</sup> had extensive GFP and absence of pSTAT5 in the MPOA, indicative of loss of Prlr. (**B**) As reported previously (**Brown et al., 2017**), AAV-cre-mediated

*Figure 6 continued on next page*

*Figure 6 continued*

deletion of Prlr from the MPOA resulted in failure of maternal behavior, and no pups from MPOA AAV-cre injection *Prlr*^lox/lox^ mice survived to day 3 of lactation. (**C**) Average daily RWA in non-pregnant state. (**D**) RWA activity before and after successful mating in mice lacking Prlr in the MPOA following AAV-cre injection (AAV-cre MPOA injection). RWA activity for each mouse is expressed as a percentage of their individual average daily levels in the non-pregnant state (control injection, n = 8, AAV-cre injection, n = 9, total daily RWA per animal shown in *Figure 6—source data 1*). (**E**) Change in RWA from non-pregnant state to early pregnancy in AAV-cre-injected and control-injected mice. Dotted line indicates mean non-pregnant RWA for each group. (**F**) RWA on day 15 pregnancy for AAV-cre-injected mice (n = 9) and controls (n = 8). (**G**) Cumulative RWA over days 16–18 pregnancy for AAV-cre-injected mice (n = 9) and controls (n = 8).

The online version of this article includes the following figure supplement(s) for figure 6:

**Source data 1.** Each line (gray = control, red = AAV medial preoptic area[ MPOA] injection of cre) represents total daily running wheel activity (RWA) from each individual mouse in the non-pregnant state (4 days before mating) and during pregnancy (days 1–18).

prolactin action in the brain breakthrough bouts of extensive RWA are possible. There was no significant difference in early lactation RWA for *Prlr*^lox/lox^/*Slc32a1*^Cre^ mice and controls (*Figure 7B*, interaction genotype × time: p=0.4394, $F_{(92, 23)} = 0.8525$; effect of genotype: p=0.2292, $F_{(1, 15)} = 1.571$). During lactation, mice spent an extensive amount of time nursing offspring and were in a negative state of energy balance, either of which may impact this change in activity. Since complete deletion of Prlr from the MPOA leads to abandonment of pups and termination of lactation, it was not possible to assess the role of Prlr in this area in the suppression of RWA during lactation.

## Discussion

In this study, we have shown that the mating-induced release of prolactin, acting through Prlr in the brain, induces a marked reduction in RWA in mice very early in their pregnancy, and this suppressive effect continues throughout pregnancy and into the first few days of lactation. It seems likely that this profound behavioral response is an important addition to a range of hormone-induced metabolic adaptations that have been characterized to facilitate a positive energy balance during pregnancy, including leptin resistance, increased appetite, and reduced energy expenditure (*Ladyman et al., 2018a*; *Ladyman et al., 2012*). From an evolutionary perspective, these can be viewed as adaptive strategies, enabling pregnant females to build up energy reserves in preparation for the pronounced metabolic demands associated with fetal growth and the subsequent lactation (*Butte and King, 2005*; *Dunsworth et al., 2012*). In our modern obesogenic environment, however, such changes have become maladaptive, contributing to excessive weight gain during pregnancy with the associated increased risk in pregnancy complications (*Yang et al., 2017*). Increasing numbers of women are coming into pregnancy already overweight or obese, and over 50% of women now exceed the Institute of Medicine (IOM) guidelines for optimal weight gain during pregnancy (*Chung et al., 2013*; *Goldstein et al., 2017*; *Voerman et al., 2019*). Indeed, obesity during pregnancy has been described as the '*most common clinical risk factor encountered in obstetric practice*' (*Krishnamoorthy et al., 2006*). As it is difficult to alter energy intake, increasing energy expenditure through exercise during pregnancy has become highly advocated as a therapeutic intervention that produces healthier gestational weight gain (*Sanabria-Martinez et al., 2015*). However, despite pregnant women being well-informed about the benefits and safety of exercise during pregnancy (*Harrison et al., 2018*), 60–80% of pregnant women do not engage in physical activity as recommended (*Amezcua-Prieto et al., 2013*; *Gaston and Vamos, 2013*; *Hesketh and Evenson, 2016*). This suggests a disconnect between knowledge and action (*Harrison et al., 2018*), and poses the question of whether there is a biological basis for decreased physical exercise during pregnancy. Indeed, women report fatigue, pregnancy symptoms, and lack of motivation all as barriers to exercise during pregnancy (*Harrison et al., 2018*). Here, we describe a previously unknown hormonal mechanism that may be contributing to the loss of motivation for exercise in pregnancy.

Three distinct approaches to deleting Prlr in specific neuronal populations each effectively prevented the initial pregnancy-induced suppression of RWA. Firstly, a broad forebrain neuron-specific deletion of Prlr using the *Prlr*^lox/lox^/*Camk2a*^Cre^ mouse blocked the effect, implicating a prolactin action in the brain, but this model was complicated by hyperprolactinemia and abnormal estrous cycles. Using the *Prlr*^lox/lox^/*Slc32a1*^Cre^ mice to provide a more targeted deletion of Prlr in GABAergic neurons also blocked the effect. Validation work showed that this genetic cross resulted in over 70% deletion of

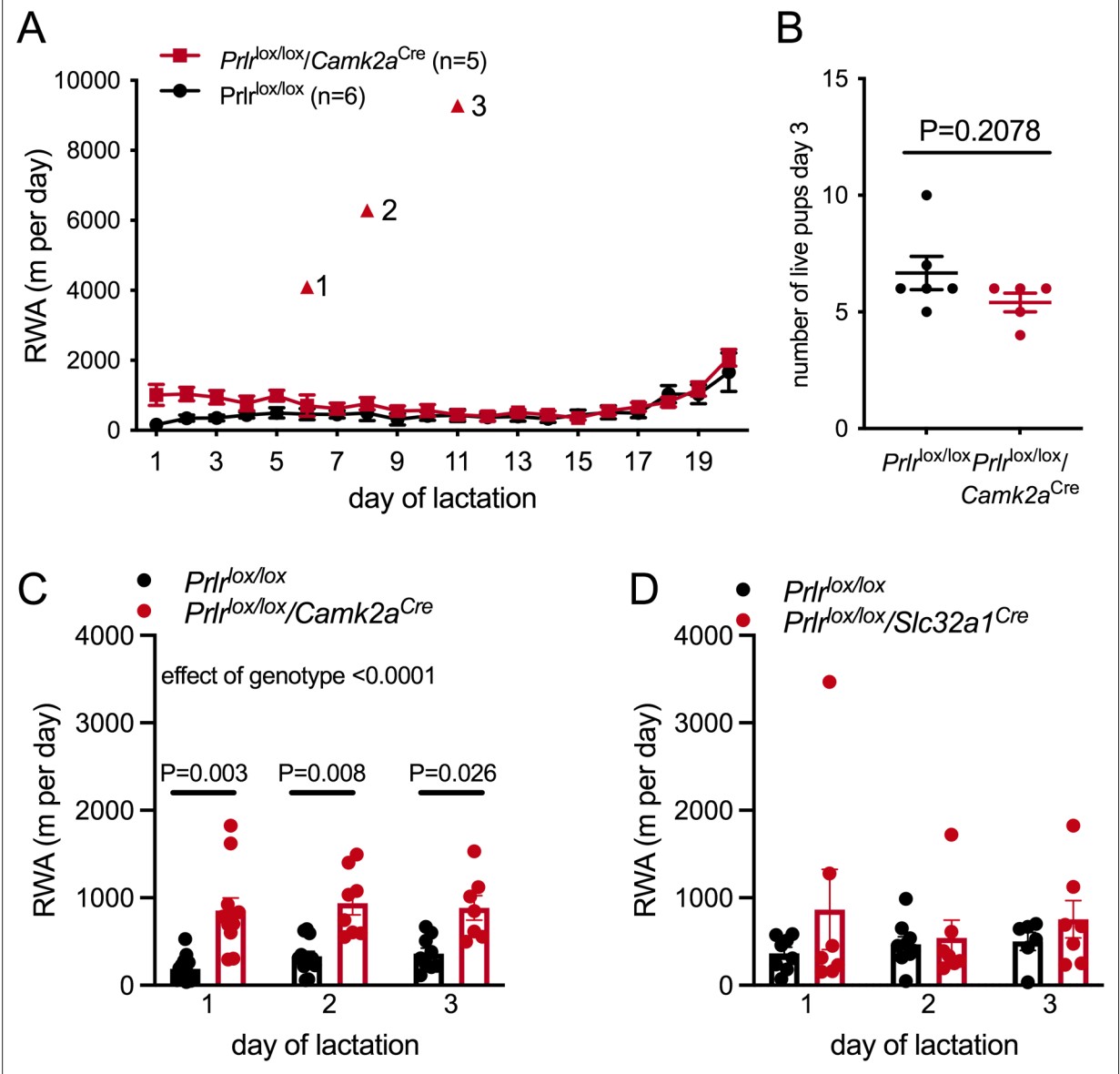

**Figure 7.** Prolactin contributes to early lactation attenuation in running wheel activity (RWA) but not through GABA neurons. (**A**) Daily RWA across lactation in a small cohort of *Prlr*^lox/lox^/*Camk2a*^Cre^ and control littermates (*Prlr*^lox/lox^). Triangles 1–3 each indicate RWA for an individual mouse on a single day of excessive RWA in mid lactation. (**B**) Number of pups alive per litter on the morning of day 3 of lactation for each dams in (**A**), no pups died across lactation, and so this graph also represents the number of pups for each litter at weaning. (**C, D**) Bars represent average daily RWA in control (*Prlr*^lox/lox^) and mice lacking Prlr in forebrain neurons (**C**) (*Prlr*^lox/lox^/*Camk2a*^Cre^, n = 13, *Prlr*^lox/lox^, n = 11, p-value from *post hoc* test between genotype at each time point) or GABA neurons (**D**) (*Prlr*^lox/lox^/*Slc32a1*^Cre^, n = 8, *Prlr*^lox/lox^, n = 9).

Prlr expression from GABAergic neurons, leaving Prlr expression in other neurons unaffected. Finally, a region-specific deletion of Prlr from the MPOA using an AAV-cre in adult mice also completely blocked the effect. Collectively, we interpret these data as implicating a population of neurons in the MPOA, at least some of which are likely to be GABAergic, in mediating this profound change in behavior in response to elevated prolactin in early pregnancy. Deletion of Prlr from arcuate nucleus kisspeptin neurons demonstrated a smaller, more subtle attenuation of the early pregnancy-induced suppression in RWA, suggesting that prolactin-sensitive kisspeptin neurons in the arcuate nucleus may make a minor contribution to the immediate reduction in RWA in early pregnancy.

After the initial decrease in RWA early in pregnancy, RWA is fairly stable until about day 11 onwards when further decreases in RWA occurs, and by late pregnancy levels are extremely low (*Ladyman et al., 2018a*). This additional decline in RWA in mid-pregnancy takes place when body weight is

starting to increase rapidly. and hence it seems likely that this further decrease in RWA is due to physical constraints caused by fetal growth, but this is yet to be investigated. Despite this overall decrease in RWA in the latter parts of pregnancy, Prlr KO models still showed attenuated suppression of RWA compared with control pregnant mice, demonstrating that an inhibitory action of prolactin on RWA is present throughout pregnancy. Interestingly, in mice with Prlr deleted in GABA neurons, this attenuated suppression of RWA was progressively less evident as pregnancy advanced, becoming similar to controls over the last few days of pregnancy. In contrast, the forebrain Prlr KO mouse and the MPOA-specific Prlr KO showed elevated levels of RWA compared to controls that remained elevated throughout pregnancy. This difference suggests the likelihood that several distinct prolactin-sensitive pathways regulate the effects of pregnancy on RWA. The early pregnancy response we originally set out to study seems to predominantly involve a GABAergic subpopulation in the MPOA. In late pregnancy, a non-GABAergic population appears to play the major contributing role. In lactation, prolactin only contributes to the suppression of RWA in the first few days while lactation is being established, and this also appears to be through a non-GABAergic population. While it is tempting to speculate that this lactation effect would be mediated by the same neuronal population as the effect in late pregnancy, mice with an MPOA-specific KO of Prlr do not establish lactation, and hence we could not evaluate RWA in early lactation in that model.

The MPOA is involved in regulating a wide range of homeostatic and behavioral functions, including parental behavior (*Kohl et al., 2017*), and in particular, prolactin action on maternal behavior (*Bridges, 2015*). We have recently shown (replicated in *Figure 6*) that prolactin action in this area is critical to the normal expression of maternal behavior in mice (*Brown et al., 2017*). Hence, the prolactin-mediated effect on RWA, described here, might be considered part of the complex adaptive role of the MPOA in promoting appropriate parental responses, triggered by the hormonal changes of pregnancy and lactation. In contrast to the total failure of maternal behavior in mice lacking Prlr in the MPOA, however, the 'forebrain Prlr deletion' mouse line were still able to care for their offspring. Despite losing only approximately 50% of Prlr in the MPOA, these animals clearly showed attenuated pregnancy-induced and early lactation-induced suppression of RWA, suggesting that either the neurons influencing RWA are only a subset of those required for the expression of maternal behavior or that there is no major overlap in these neuronal pathways. The MPOA has previously been linked to RWA, with the increased running in response to estradiol treatment apparently mediated through estrogen receptor alpha (ERα) in the MPOA (*Grigsby et al., 2020*; *King, 1979*; *Wei et al., 2018*). The neuronal cell types mediating the RWA response to estradiol have not been elucidated, but an estrogen-responsive MPOA circuit projecting to the VTA has been shown to be critical for maternal behavior (*Wei et al., 2018*; *Fang et al., 2018*), and at least some of these ERα-expressing neurons in the MPOA are GABAergic (*Herbison, 1997*). It is possible that estradiol and prolactin may be targeting similar populations of neurons to impact this behavior. Given the strong evidence linking VTA dopamine projections with the rewarding aspects of RWA, this remains a strong candidate for mediating the effect of prolactin on RWA. Within the MPOA, there is extensive heterogeneity in neuronal cell types that express the Prlr, even amongst GABAergic subtypes (*Moffitt et al., 2018*), and further work will be required to determine the specific cell type involved in this action of prolactin.

The fact that the very low RWA seen in late pregnancy persists throughout lactation, when the physical limitations imposed by pregnancy have resolved, points more to a hormonal mechanism being involved. Interestingly, our data suggest that prolactin is only partially involved in the ongoing suppression seen during lactation, and other factors must also be involved. A possible candidate is progesterone, which is known to decrease RWA in rodents through opposing the positive influence of estradiol (*Lightfoot, 2008*). In mice, progesterone begins to increase around 48 hr following detection of a copulatory plug and then is high-throughout pregnancy until around 24 hr before parturition (*Barkley et al., 1979*; *Murr et al., 1974*). After birth, progesterone secretion is reinitiated in rodents due to prolactin-induced rescue of the corpus luteum following a postpartum ovulation. While there is evidence from progesterone receptor knockout rats that progesterone interactions with this receptor are not required for regulating RWA during the estrous cycle (*Kubota et al., 2016*), it seems possible that the very high levels of progesterone seen during pregnancy and lactation are facilitating the prolactin-induced suppression of RWA as pregnancy advances and also contributing during lactation. An alternative possibility is that the ongoing suppression of RWA in lactation involves a different mechanism to that seen in pregnancy. The somatosensory inputs provided by the suckling stimulus

initiated at the nipple are known to influence motor functions in the mother, promoting a quiescent, upright nursing posture (kyphosis) to allow pups to readily access and suckle at the nipples (*Stern and Lonstein, 2001*). Conceivably, these same neural inputs may suppress motivation to engage in RWA.

Overall, the data provide novel and convincing evidence of a prolactin-mediated adaptive change in RWA during early pregnancy in mice that continues to contribute, along with other factors, to the suppression of RWA as pregnancy progresses. Collectively, the engagement of multiple signals to drive a reduction in RWA in pregnancy and lactation suggests that there must be some biological advantage to successful reproduction of inducing this profound change in behavior. Under the conditions we housed our animals in, there was not any catastrophic failure in pregnancy outcomes associated with transgenic mouse lines that engaged in more RWA during pregnancy. Under less optimal conditions, however, a failure to show this adaptive response might have more serious consequences. Whether a prolactin-mediated adaptive change in voluntary physical exercise also occurs in women during pregnancy requires attention. Prolactin, and its closely related placental analogue, placental lactogen are both highly regulated hormones during human pregnancy, among the top 1% of proteins increasing in the blood at this time (*Aghaeepour et al., 2018*; *Romero et al., 2017*). These changes occur progressively throughout human pregnancy, as opposed to being an immediate mating-induced increase as seen in rodents, but are still likely to be sufficiently elevated in the blood to be having effects in the brain by the end of the first trimester. It seems probable that these elevated levels of lactogenic hormones in women are contributing to the broad range of adaptations that are occurring during pregnancy (*Napso et al., 2018*; *Grattan and LeTissier, 2015*). As discussed above, however, in the current environment some of these changes may have become maladaptive and potentially compromise healthy behaviors. Such a possibility needs to be seriously considered in providing obstetric advice to pregnant women.

## Materials and methods

**Key resources table**

| Reagent type (species) or resource | Designation | Source or reference | Identifiers | Additional information |
|---|---|---|---|---|
| Strain, strain background (*Mus musculus*, female and male) | C57BL/6J | Jackson Laboratory | Stock #:000664; RRID:IMSR_JAX:000664 | |
| Genetic reagent (*M. musculus*) | Prlr^lox/lox | PMID:27581458 | MGI:6196142 | Prof. Dave Grattan (University of Otago, New Zealand) |
| Genetic reagent (*M. musculus*) | Camk2a^Cre | PMID:11668676 | RRID:IMSR_EM:01153 | PMID:11668676 |
| Genetic reagent (*M. musculus*) | Slc32a1^Cre | Jackson Laboratory | Stock #:028862; RRID:IMSR_JAX:028862 | PMID:21745644 |
| Genetic reagent (*M. musculus*) | Kiss1^Cre | PMID:21149719 | MGI:4878876 | Prof. Ulrich Boehm (Saarland University Germany) |
| Antibody | GFP, polyclonal rabbit | Life Technologies | Cat #:A6455; RRID:AB_221570 | (1:20,000) |
| Antibody | Phospho STAT5, polyclonal rabbit | Cell Signaling Technology | Cat #:9351; RRID:AB_2491009 | (1:1000) |
| Sequence-based reagent | Slc32a-probe, Mm-Slc32a1-C2 | Advanced Cell Diagnostic | Cat #:319191_C2 | Targeting the nucleotide sequence in the region 894–2037 of NM_009508.2 |

*Continued on next page*

*Continued*

| Reagent type (species) or resource | Designation | Source or reference | Identifiers | Additional information |
|---|---|---|---|---|
| Sequence-based reagent | Prlr probe, Mm-Prlr-01 | Advanced Cell Diagnostic | Cat #:ADV588621 | Targeting the nucleotide sequence in the region 1107–2147 of NM_011169.5 |
| Recombinant DNA reagent | AAV/DJ-CMV-mCherry-iCre | Vector Biosystems | Cat #:VB7600 | |
| Recombinant DNA reagent | AAV/DJ-CMV-mCherry | Vector Biosystems | Cat #:VB7777 | |
| Peptide, recombinant protein | Prolactin, ovine | Sigma-Aldrich | Cat #:L6520 | |
| Peptide, recombinant protein | Prolactin, ovine | National Hormone and Pituitary Program, NIDDK | Cat #:AFP-10692C | |
| Commercial assay or kit | RNAscope 2.5HD Duplex assay | Advanced Cell Diagnostic | Cat #:ADV322430 | |

## Animals

Female mice starting at age 8–12 weeks were housed in a temperature- and lighting-controlled environment (22 ± 1° C, 12 hr light:12 hr dark, lights on at 0700 hr) and allowed access to food and water ad libitum. When required, mice were handled daily to monitor estrous cyclicity. All experimental protocols were approved by the University of Otago Animal Ethics Committee. Groups of C57BL/6J mice were used for characterization of activity during pregnancy and to investigate the effects of acute prolactin on activity. *Prlr*<sup>lox/lox</sup> mice were generated as previously described (**Brown et al., 2016**). CamKII-alpha cre (*Camk2a*<sup>Cre</sup>) mice (**Casanova et al., 2001**), *Slc32a1*<sup>Cre</sup> (**Vong et al., 2011**; Jackson lab stock # 028862), and *Kiss1*<sup>Cre</sup> (**Mayer et al., 2010**) were crossed with *Prlr*<sup>lox/lox</sup> mice to generate mouse lines with deletion of Prlr in various neuronal populations.

## Prolactin treatment

Exogenous prolactin was either ovine prolactin (Sigma-Aldrich) dissolved in saline or ovine prolactin (obtained from Dr. A. F. Parlow, National Hormone and Pituitary Program, National Institute of Diabetes and Digestive and Kidney Diseases, Torrance, CA) dissolved in PBS/130 mM NaCl (pH 8) to inject at a dose of 5 mg/kg (i.p.). Vehicle was either saline or PBS/130 mM NaCl (pH 8). For studies with a counterbalanced design, mice underwent both prolactin injection and vehicle injection, with at least one estrous cycle between treatment days (or 4 days for males). For the effect of acute prolactin on RWA and home cage ambulation, injections were given 30 min before the onset of the dark period. All injections were given i.p., and all mice were habituated to handling. All injections in female mice were carried out in the metestrous phase of the estrous cycle.

## Running wheel activity

All mice were housed individually for the assessment of RWA. For assessment of effect of running wheel type on pregnancy-induced changes in RWA, mice with access to an upright wheel were housed in Promethion metabolic and behavioral phenotyping cages (Sable Systems International) for approximately 2 weeks, moved to a cage with a stud male until the day a copulatory plug was detected, at which point they were returned to their Promethion metabolic and behavioral phenotyping cages. For comparison between wheel types, mice in the upright wheel group consisted of C57BL/6J mice and control *Prlr*<sup>lox/lox</sup> mice. Mice in the saucer wheel group consisted of *Prlr*<sup>lox/lox</sup> mice and were housed with wheel access for at least 3 weeks in the virgin state then similarly treated as those in the upright wheel group. For upright wheel group, running wheel data was collected and processed using MetaScreen

and Expedata programs. For saucer wheel group, wheel data was collected and processed using Wheel Manager Software (version 2.03, MED Associates, Inc).

Studies using *Prlr*<sup>lox/lox</sup>/*Camk2a*<sup>Cre</sup>, *Prlr*<sup>lox/lox</sup>/*Slc32a1*<sup>Cre</sup>, and *Prlr*<sup>lox/lox</sup>/*Kiss1*<sup>Cre</sup> (and their controls) recorded running activity from upright wheels in metabolic and behavioral monitoring cages (Promethion, Sable Systems International), while studies with *Prlr*<sup>lox/lox</sup> mice with AAV-cre injected into the MPOA (and their controls) used wireless low-profile 'saucer or disc type' running wheels (Med Associates Inc, Vermont, USA). For all studies, mice were habituated to the running wheels in the virgin state (approximately 2 weeks for upright wheels and at least 3 weeks for low-profile wheels). For pregnancy studies, mice were removed from their home cages and housed with a stud male (C57BL/6J ) mouse until detection of a plug indicating successful mating. Females were then returned to home cages with wheels and were not disturbed till day 3 or 4 of lactation when the number of offspring was assessed. Pregnancy-induced changes in RWA were examined in two cohorts of *Prlr*<sup>lox/lox</sup>/*Camk2a*<sup>Cre</sup> and control mice at different times, and data was pooled as results from the first cohort were repeated in the second cohort. In one cohort, RWA was also monitored through lactation. For all other transgenic mouse lines (and their littermate controls), the experiment was only performed on one cohort of mice. For the second cohort of *Prlr*<sup>lox/lox</sup>/*Camk2a*<sup>Cre</sup> and the cohort of *Prlr*<sup>lox/lox</sup>/*Slc32a1*<sup>Cre</sup>, RWA was only monitored for the first 4 days of lactation. Data from the virgin and pregnant state were only included in analysis if mice successfully became pregnant and gave birth.

Our data *Figure 1* and *Ladyman et al., 2018a* demonstrate that while there is a wide variation in stable, total daily RWA between different mice, pregnancy induces an approximately 50% decrease in RWA as soon pregnancy is achieved independent of whether a mouse is a 'high' or 'low' runner. Due to the varying range of stable running levels in individual mice, to assess the pregnancy-induced change in behavior we analyzed the percentage change in RWA for each individual mouse as opposed to using total distance per day for each mouse (although this data is provided in source data for each transgenic line). Non-pregnant RWA was calculated for the 4 days prior to mating as a percentage of the average daily RWA over at least a week following running wheel habituation. RWA activity for each day of pregnancy was also calculated in a similar manner as a percentage of average daily RWA over at least a week.

## Acute effects of prolactin on RWA

Female and male C57BL/6J  mice (n = 12 per sex) were housed individually with running wheels for at least 3 weeks. Estrous cycle was monitored daily in females and treatment (prolactin or vehicle) was injected on the metestrous phase of the estrous cycle. Mice were administered in a randomized, alternate order such that half received 5 mg/kg injections (i.p.) of vehicle or prolactin first and on the next metestrus the same animals received the alternate treatment. Males were first injected with saline or prolactin and 4 days later received the alternate treatment. Both groups received injections approximately 30 min before the onset of the dark phase. This experiment was carried out in one cohort of mice for each sex.

## Acute effects of prolactin on distance traveled in novel environments

All experiments examining acute effects of prolactin on various physical activity measures in female mice was performed in one cohort of mice, although the experiments for different measures used different cohorts of mice. Mice were housed in groups of four or five per cage. Female C57BL/6J  mice (n = 16) were monitored for stage of estrous cycle, and on metestrus they were injected i.p. with either vehicle or prolactin (5 mg/kg) approximately 4 hr into the dark phase of the light/dark cycle. One hour later mice were placed in an open field box and recorded using TopScan software (CleverSys, Inc, VA) for the next 30 min. OF testing was carried out under sodium lighting to allow video recording but perceived as 'dark' to rodents. After completing a full estrous cycle, on the next metestrus, mice were similarly injected with vehicle or prolactin (5 mg/kg) approximately 4 hr into the dark phase of the light/dark cycle and then an hour later mice were placed in an EPM. During the EPM, mice were exposed to white light for the duration of the test. The activity of the mouse was recorded using TopScan software and both OF and EPM tests were analyzed for distance traveled during the time period (OF: 30 min; EPM: 5 min) using TopScan software.

## Acute effects of prolactin on activity in familiar environment

Female C57BL/6J mice were individually housed in Promethion cages and underwent 2 weeks of handling (removal from cages) and estrous cycle monitoring prior to experiment. On metestrus, mice received an injection of prolactin (5 mg/kg) or vehicle approximately 30 min before the start of the dark phase of the light/dark cycle. On the metestrus of the following estrous cycle, mice received the alternative treatment as described above. Data was collected from Promethion cages using Metascreen software and processed using Expedata software. End points analyzed included total ambulation distance, total infrared beam breaks, and non-ambulatory movements.

## Measure of anxiety-like behavior during pregnancy

Individually housed, pregnant $Prlr^{lox/lox}/Camk2a^{Cre}$, $Prlr^{lox/lox}/Slc32a1^{Cre}$, and $Prlr^{lox/lox}$ mice underwent an EPM test on day 15 or 16 of pregnancy similar to that described above but with no prior treatment. The activity of the mouse was recorded and analyzed using TopScan software. At completion of the EPM, mice were killed and the uterus, including contents, was weighed and fetal number was recorded.

## Stereotaxic injections of AAV

Female $Prlr^{lox/lox}$ mice (8–12 weeks of age, housed in groups of four or five) were anesthetized with isoflurane and placed in a stereotaxic apparatus. Mice received bilateral 0.8 µl injections of AAV/DJ-CMV-mCherry-iCre ($1.8 \times 10^{13}$, Vector Biosystems) for gene deletion group or AAV/DJ-CMV-mCherry ($3.7 \times 10^{13}$, Vector Biosystems) for control group into the MPOA (coordinates were 0.07 anterior to Bregma, 0.3 mm lateral to midline, and depth from top of the brain was adjusted by body weight: < 22 g 4.7 mm depth, > 22 g 4.9 mm). Injections were given at a rate of 80 nl/min, and the syringes were left in situ for 3 min before and 10 min following completion of injection. Mice were left to recover for 1 week, then housed individually with running wheels for at least 3 weeks prior to mating. Only mice that showed GFP staining in the MPOA, successfully went through pregnancy, and abandoned their litters as previously described (*Brown et al., 2017*) were included in the MPOA-specific deletion of Prlr group. For the control group, only mice that underwent surgery for control virus injection, showed no GFP staining in the MPOA, and successfully went through pregnancy were included (inclusion/exclusion for the control group was not influenced by successful nurturing of offspring, although no mice in this group abandoned their pups).

## Immunohistochemistry

$Prlr^{lox/lox}/Camk2a^{Cre}$, $Prlr^{lox/lox}/Slc32a1^{Cre}$, and $Prlr^{lox/lox}/Kiss1^{Cre}$ mice have been previously characterized (*Brown et al., 2016*; *Brown et al., 2017*; *Brown et al., 2019*). To confirm the deletion of Prlr in this study, immunohistochemistry for prolactin-induced pSTAT5 was used for $Prlr^{lox/lox}/Camk2a^{Cre}$ and $Prlr^{lox/lox}/Slc32a1^{Cre}$ lines, while immunohistochemistry for GFP, which is indicative of cre-mediated recombination of the Prlr gene in this $Prlr^{lox/lox}$ mouse line, was used for the $Prlr^{lox/lox}/Kiss1^{Cre}$ line. Both prolactin-induced pSTAT5 and GFP immunohistochemistry were used to demonstrate deletion of Prlr in MPOA AAV-cre-injected $Prlr^{lox/lox}$ mice. For pSTAT5 immunohistochemistry, mice received an i.p. injection of prolactin (5 mg/kg) 45 min before perfusion. Mice were anesthetized with sodium pentobarbital then perfused transcardially with 4% paraformaldehyde. Brains were removed and processed for immunohistochemistry for GFP or pSTAT5, as previously described (*Brown et al., 2016*). Briefly, for pSTAT5 immunohistochemistry coronal brain sections (30 µm) underwent an antigen retrieval procedure consisting of incubation in 0.01 mM Tris (pH 10) at 90° C for 5 min, then were left to cool for 10 min. Sections were incubated in rabbit anti-phospho STAT5 antibody (dilution 1:1000, polyclonal rabbit anti-phospho-STAT5, Tyr 694, #9351, Cell Signaling Technology, Beverly, MA) for 48 hr. Following this, sections were incubated in biotinylated goat anti-rabbit IgG (dilution 1:300, BA-1000, Vector Laboratories, Inc, Burlingame, CA) for 90 min and then incubated in Vector Elite avidin-biotin-HRP complex (dilution 1:100) for 60 min and labelling was visualized with nickel diaminobenzidine tetrahydrochloride using glucose oxidase to create a black, nuclear precipitate. For GFP, immunohistochemistry sections were incubated in anti-GFP antibody (dilution 1:20,000, polyclonal rabbit-anti GFP, A6455, Life Technologies, Grand Island, NY) for 48 hr at 4 °C. Sections were then treated as for pSTAT5 immunohistochemistry. Chromagen immunohistochemistry was examined using a light microscope at either

10× or 20× , and numbers of positively stained cells were counted in either two sections (MPOA, PVN) or three sections (ARC, VMN) per mice and anatomically matched between mice.

## RNAscope in situ hybridization

*Prlr* mRNA in MPOA GABA neurons was assessed by RNAscope in situ hybridization in *Prlr*^lox/lox^/*Cam-k2a*^Cre^ and *Prlr*^lox/lox^/*Slc32a1*^Cre^ mice and their respective littermate controls. Non-pregnant female (8–10 weeks of age, group housed) mice were perfused on random days of the estrous cycle with 2% paraformaldehyde and sections (14 µm) through the POA sliced in a cryostat and float-mounted onto SuperFrost plus slides (Thermo Fisher Scientific NZ Ltd, North Shore City, New Zealand), air dried for 1 hr at room temperature, and stored at –20° C till further processing. *Prlr* and *Slc32a1*-positive cells were detected using specific probes (Prlr: Mm-Prlr-01, targeting the nucleotide sequence in the region 1107–2147 of NM_011169.5, Cat # ADV588621, vGAT: Mm-Slc32a1-C2, targeting the nucleotide sequence in the region 894–2037 of NM_009508.2, Cat # 319191_C2) in an RNAscope 2.5HD Duplex assay in accordance with the manufacturer's instructions with minor modifications (Advanced Cell Diagnostics, Hayward, CA). Briefly, sections were thawed at 55° C and post-fixed for 3 min in 2% paraformaldehyde. After incubation in RNAscope hydrogen peroxidase solution to block endogenous peroxidases, tissue was immersed in 100% ethanol then air dried. Tissue was permeabilized by incubation in RNAscope protease plus solution for 30 min at 40° C then incubated in a mixture of the *Prlr* probe and *Slc32a1* probe (1:500 ratio). This was followed by a series of amplification incubation steps between which sections were washed twice with the provided washing buffer. After amplification steps 1–6, positive hybridization for *Slc32a1* was detected by incubation with provided detection reagents (Fast-RED B:Fast-RED A, 1:60) for 10 min at room temperature. Following another four amplification steps interspersed with washes, positive hybridization for *Prlr* was detected by incubation with provided detection reagents (Green-B:Green A, 1:50) for 10 min. Slides were then dried and coverslipped using VectaMount mounting medium (Vector Laboratories). Images were taken using an AX70 Provi light microscope (Olympus, Tokyo, Japan) and attached Spot RT digital camera (Spot Imaging, Sterling Heights, MI), and number of cells within the POA that displayed positive hybridization for both probes, as determined using Advanced Cell Diagnostic's RNAscope scoring guidelines, was assessed in two sections per animal. Sections were anatomically matched between animals. Separate RNAscope in situ hybridization runs were carried out for each transgenic mouse line and hence they were analyzed separately.

## Statistical analysis

Statistical analyses were carried out with GraphPad Prism software. All group sizes refer to biological replications. $p < 0.05$ was considered statistically significant, and all data are presented as the mean ± SEM. ROUT outlier test was used to determine any outliers, and outliers were removed from the data group. Student's t-test was used to analyze daily RWA activity with different wheel types, distance traveled in EPM and OFT, virgin daily RWA, pSTAT5-positive cell number, virgin body weight, uterus weight, and fetus number. Paired Student's t-test was used to analyze EE and ambulation in early pregnancy and total overall RWA, fine movement, and time spent engaging in different behaviors following prolactin or vehicle treatment. Two-way ANOVA followed by Sidak's multiple comparisons test was used to analyze the change in RWA from non-pregnant (4 days prior to mating) to early pregnancy (days 1–3 of pregnancy). Repeated measures ANOVA followed by Sidak's multiple comparisons test was used to analyze prolactin effects on RWA, ambulation, XY beam breaks, and fine movement in virgin state. One-way ANOVA was used to analyze daily EE and ambulation during the different weeks of pregnancy. Mixed effects analysis followed by Sidak's multiple comparisons test was used to analyze changes in daily RWA, body weight, and food intake across pregnancy. Mann–Whitney test was used to analyze estrous cycle data, pup number during early lactation, GFP-positive cell number, and number of *Prlr*-positive, *Slc32a1*-positive cells determined by RNAscope in situ hybridization.

## Acknowledgements

We thank Pene Knowles for genotyping mice, Chantelle Murrell for assistance with immunohistochemistry quantification, and Dr Holly Phillipps for assistance with the RNAscope in situ hybridization procedure. This work was supported by a Health Research Council of New Zealand Programme Grant

14-568 (to DRG) and University of Otago School of Biomedical Sciences/Dunedin School of Medicine research grant funding (to SRL).

## Additional information

### Funding

| Funder | Grant reference number | Author |
|---|---|---|
| Health Research Council of New Zealand | 14-568 | David R Grattan |
| University of Otago | | Sharon R Ladyman |
| University of Otago School of Biomedical Sciences/ Dunedin School of Medicine | | Sharon R Ladyman |

The funders had no role in study design, data collection and interpretation, or the decision to submit the work for publication.

### Author contributions

Sharon R Ladyman, Conceptualization, Data curation, Formal analysis, Funding acquisition, Investigation, Methodology, Project administration, Supervision, Writing - original draft; Kirsten M Carter, Matt L Gillett, Data curation, Formal analysis, Investigation; Zin Khant Aung, Investigation; David R Grattan, Conceptualization, Funding acquisition, Supervision, Writing - review and editing

### Author ORCIDs

Sharon R Ladyman https://orcid.org/0000-0003-3578-6763
Matt L Gillett https://orcid.org/0000-0001-8167-4354
Zin Khant Aung https://orcid.org/0000-0002-5121-2770
David R Grattan https://orcid.org/0000-0001-5606-2559

### Ethics

This study was performed in strict accordance with the Animal Welfare Act (1999) New Zealand. All experimental protocols were approved by the University of Otago Animal Ethics Committee (Animal Use Protocol 36-17).

### Decision letter and Author response

Decision letter https://doi.org/10.7554/eLife.62260.sa1
Author response https://doi.org/10.7554/eLife.62260.sa2

## Additional files

### Supplementary files

• Transparent reporting form
• Source data 1. Running wheel activity levels across pregnancy.

### Data availability

All data generated or analysed during this study are included in the manuscript and supporting files. Source data files have been provided for Figures 4-6.

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
