## [Decision Letter]

**Acceptance summary:**

This manuscript provides new insights into the management of energy during pregnancy by determining that the hormone prolactin acts within the brain to immediately reduce activity when pregnancy occurs. Previously prolactin was known for its role in maintaining the corpus luteum, required for pregnancy, and in the maternal behavior and lactation associated with the postpartum period, but the findings here expands its role as well as inform about the mechanistic origins of this previously over looked behavioral response.

**Decision letter after peer review:**

Thank you for submitting your article "A reduction in voluntary physical activity during pregnancy in mice is mediated by prolactin" for consideration by *eLife*. Your article has been reviewed by 3 peer reviewers, including Peg McCarthy as Reviewing Editor and Reviewer #1, and the evaluation has been overseen by Kate Wassum as the Senior Editor. The following individual involved in review of your submission has agreed to reveal their identity: Stephanie Padilla (Reviewer #2).

The reviewers have discussed the reviews with one another and the Reviewing Editor has drafted this decision to help you prepare a revised submission.

Summary:

Pregnancy generally occurs in a state of positive energy balance, which can be achieved, in part, by less physical activity in the form of voluntary exercise. In this manuscript Ladyman et al., use a mouse model to investigate a neural mechanism underlying decreased voluntary exercise during early pregnancy. They show that mice are less active (voluntary exercise and home-cage activity) during pregnancy; coincident with increased levels of the hormone prolactin. Next they demonstrate that an acute injection of prolactin is sufficient to decrease WRA during the dark phase, in females but not males. Finally, using a series of genetic crosses to knock-out prolactin receptor in various cell types within the brain, they investigate the necessity of prolactin signaling to attenuate voluntary WRA in early pregnancy. The authors conclude that prolactin signaling in the MPOA is a key site by which prolactin mediates an influence on WRA.

This report provides a plausible, albeit limited, mechanistic explanation for the rapid and dramatic decrease in locomotion seen in mice following initiation of pregnancy. The purpose of the two daily surges in prolactin following successful mating has always been deemed to be the rescue and continued function of the corpus luteum, thereby maintaining the pregnancy. But the work here demonstrates a behavioral role for the prolactin as well by reducing locomotion, presumably for purposes of reducing metabolic burn in anticipation of the pending increased caloric demands.

The paper shows interesting and new findings and potentially contribute with the field by describing a new role of prolactin in the brain controlling a specific behavior. However, some points still need to be better addressed by the authors in order to strengthen their hypothesis:

Essential revisions:

1) The validation data of the manuscript needs to be improved. Although the authors showed that prolactin receptor deletion from VGAT positive neurons reduced the number of pSTAT5 cells in the MPOA and ARC, a lot of prolactin-induced pSTAT5 cells could still be observed. These remaining cells may not express VGAT or the deletion may affect only part of the VGAT-positive neurons. To clarify this point, a co-localization study should be performed to indicate whether the remaining pSTAT5 cells in the knockout animals express do or do not VGAT.

2) In all mouse models shown (CamK-Cre, vGat-Cre and AAV-Cre MPOA), there are some mice that showed a robust reduction in voluntary physical activity during pregnancy, whereas other animals show no reduction or even an increase in this behavior. This variability precludes the existence of a statistical significance. Or it indicates that the prolactin receptor deletion may not be equally effective in all animals. It is important that the authors discuss this variability that could be caused by several factors: a) insufficient PRLR deletion in the non-responders; b) simple biological variability; c) prolactin action is not relevant to regulate this behavior in a subset of animals. Especially regarding this last hypothesis, did the authors identify an aspect that differentiates responsive from non-responsive mice (e.g., animals that already show low activity before mating; number of pups at birth; low secretion of prolactin during pregnancy, etc.)? When analyzing the data individually by animals, it appears that the central action of prolactin regulating physical activity is not relevant for part of the animals, reinforcing the comment that the authors seem to overestimate their findings. Of note, in other behaviors, like maternal behavior, central prolactin action is necessary in 100% of animals. This does not seem to be the case for voluntary physical activity during pregnancy.

3) The reviewers agree that a strong connection between prolactin action and GABA neurons specifically in the POA was not made and that those claims need to be tempered or bolstered by additional experiments. Even though several mouse lineages were used, the authors did not provide a conclusive model that explain how prolactin action on GABAergic neurons of the preoptic area regulates the voluntary physical activity during pregnancy. In the Discussion, they postulate a possible regulation of dopamine secretion and consequently the rewarding aspect of the wheel running, but more experiments in this aspect are necessary to support this claim or these comments need to be clarified as being speculative. There are many additional questions that could be answered or speculated upon: Is the prolactin effect stimulatory or inhibitory in these cells? Is this a mono- or polysynaptic circuit that controls VTA dopamine neurons? How prolactin interacts with estrogen receptor α in the MPOA (the authors provided reference that ER-α cells in this area regulates wheel running)? How the neurocircuit that controls maternal behavior is related to the neurocircuit that controls voluntary activity (is there a overlap or they are different neurons)? Thus, additional experiments to describe the exact mechanism involved in the prolactin`s effects on voluntary activity seems to necessary otherwise this paper is mostly observational and it is actually not quite surprising that preoptic area mediates behavioral aspects controlled by prolactin (e.g., prolactin effects on maternal behavior is mediated by this area, so other behavioral modifications during the pregnancy/lactation cycle may also be part of these same neurocircuit).

4) In 2018, the authors published a paper describing reduction in voluntary physical activity in mice [PMID: 29738792]. In that study, the authors showed reduced running wheel activity, like the present study, but differently than here they showed normal spontaneous ambulatory activity inside the cages. In the present study, the authors showed reduced daily ambulation (Figure 1F). The authors did not discuss the possible explanations for these divergent findings and this is important to differentiate the behavior in running wheels (rewarding task) with simple ambulatory movements (possible exploratory behavior).

5) Although there is no doubt that deletion of prolactin receptor causes the maintenance of a higher running wheel activity during pregnancy, this behavior is still reduced during mid and late pregnancy. Thus, this behavior is not completely dependent on prolactin action. Although the authors discuss this point, the tone in many parts of the manuscript (title, abstract, results, conclusion) is that this behavior relies completely on prolactin signaling. Thus, it is important to say clearly that although prolactin regulates this behavior, there is likely other factors involved (not only in the Discussion).

6) Continuing the previous comment, the figures emphasize early pregnancy (which is when the authors found the most robust findings), but neglected mid and late pregnancy (these data are not shown in the figures as early pregnancy). Thus, the complete information about the changes in all pregnancy periods are important, especially because it is clear that prolactin receptor knockout mice still present reductions in this behavior, even though in a lesser extend compared to control animals.

7) Based on the results of this manuscript, I would expect estrous-cycle associated prolactin fluctuations to impact WRA. Given this, it is surprising that WRA is increased on proestrus, when circulating prolactin levels are high. Can the authors comment on this? Related: how do circulating levels of prolactin compare on proestrus versus early pregnancy in mice?

8) Throughout the manuscript the authors use the term "virgin" to refer to non-pregnant animals. Virginity is a state of never having copulated and can be entirely independent of pregnancy and so is not the correct descriptor here. Moreover, it's use suggests that the reduction in locomotion only occurs in the first pregnancy following a "loss" of virginity. Please replace this term with either "non-pregnant", or "cycling". It also raises the question, is the same degree of reduction in RWA seen in multiparous females? Or is this phenomenon limited to the first pregnancy, which would change the interpretation considerably.

9) Please include full statistical reporting (e.g., F, t statistics, degrees of freedom etc.) in the main manuscript.

---

## [Author Response]

Essential revisions:1) The validation data of the manuscript needs to be improved. Although the authors showed that prolactin receptor deletion from VGAT positive neurons reduced the number of pSTAT5 cells in the MPOA and ARC, a lot of prolactin-induced pSTAT5 cells could still be observed. These remaining cells may not express VGAT or the deletion may affect only part of the VGAT-positive neurons. To clarify this point, a co-localization study should be performed to indicate whether the remaining pSTAT5 cells in the knockout animals express do or do not VGAT.

We acknowledge this important point. We have previously reported that vGAT-positive neurons make up about 50% of the Prlr-positive neurons in the MPOA, and our present data support that observation. We have now added new RNAScope validation data to assess the degree of Prlr knockout in VGat-positive neurons in both the CKC- and VGAT-Prlr KO lines. These data have been added to figure 4 supplementary data (Figure 4 Suppl. 1B and Figure 4 Suppl. 2B), and show that the vast majority of vGat positive cells have lost Prlr expression in both of these models, and that the majority of the remaining Prlr expression is in vGat-negative neurons. This is particularly interesting for the CKC-line, which we originally included to provide a global Prlr knockout in neurons, but in the MPOA, we found that it only targets about 50% of the Prlr-expressing neurons (likely with considerable overlap with the vGat line). We have also expanded our discussion on these transgenic mouse lines, adding citations to papers containing additional validation studies characterising these lines (lines 210-219, 284-291).

2) In all mouse models shown (CamK-Cre, vGat-Cre and AAV-Cre MPOA), there are some mice that showed a robust reduction in voluntary physical activity during pregnancy, whereas other animals show no reduction or even an increase in this behavior. This variability precludes the existence of a statistical significance. Or it indicates that the prolactin receptor deletion may not be equally effective in all animals. It is important that the authors discuss this variability that could be caused by several factors: a) insufficient PRLR deletion in the non-responders; b) simple biological variability; c) prolactin action is not relevant to regulate this behavior in a subset of animals. Especially regarding this last hypothesis, did the authors identify an aspect that differentiates responsive from non-responsive mice (e.g., animals that already show low activity before mating; number of pups at birth; low secretion of prolactin during pregnancy, etc.)? When analyzing the data individually by animals, it appears that the central action of prolactin regulating physical activity is not relevant for part of the animals, reinforcing the comment that the authors seem to overestimate their findings. Of note, in other behaviors, like maternal behavior, central prolactin action is necessary in 100% of animals. This does not seem to be the case for voluntary physical activity during pregnancy.

As with any biological result there will always a degree of variation, and factors such as stress (which can acutely increase running wheel activity) cannot be ruled out as things that may influence an individual RWA measurement on any particular day. Basal levels of RWA are very variable between individuals, even in inbred mouse strains such C57BL6/J, but do seem to be quite consistent within each individual (leading some groups to pre-screen to select “high runners” or “low runners” for studies of this type PMID:9670598, PMID: 30071007). We did not try to pre-select for any particular behaviour, but rather attempted to deal with the biological variation through using appropriate group sizes and using a longitudinal study design comparing the pregnancy-induced change in RWA levels in each mouse individually as a percentage of its pre-pregnancy levels. We have not focused on absolute RWA distance, since we were specifically interested in the early pregnancy-induced change in behaviour within each individual. We have increased our use of individual data points in our figures to highlight these pregnancy-induced changes. As shown in figure 1D, despite the wide variability in absolute levels of running, and the potential for confounding factors to influence running over multiple testing days across the experiment, only 3 mice out of more than 60 demonstrated an increase in RWA in pregnancy. Out of our various control groups for the transgenic mouse studies, we only observed 2 animals that did not show a decrease in RWA% after successful mating. For a behaviour response, we have found this observation to be quite robust.

In response to this comment, we have assessed the possibility that additional variation has been introduced by “insufficient Prlr deletion” in some animals (Figure 4 supplementary 1 and 2). We did find that Prlr deletion was never 100% of the targeted population, but the levels of expression found after the various modes of Prlr deletion were actually remarkably consistent. Hence, we do not consider that variable degree of KO is a significant contributing factor to the variation in behavioral response. Nevertheless, we do agree that deletion of Prlr from the right subpopulation of neurons was important, and that not all of our manipulations have fully targeted that population. The most effective manipulation on the behaviour was the AAV-Cre mediated deletion of Prlr from the MPOA, and this also was most effective in terms of loss of functional responses to Prlr, targeting all of the heterogeneous populations of Prlr expressing neurons in this region.

With respect to the specific question about whether prolactin action is “not relevant in some animals”, we do not think this is supported by the data. As mentioned above, the pregnancy-induced suppression of RWA was robust and was observed in over 95% of control animals. Perhaps this is highlighted best in the source data provided with figure 4. We probably have insufficient “non-responders” to make strong conclusions, but we could not discern any aspect that differentiated these mice from the majority of mice that showed the characteristic early pregnancy response (for example assessing activity levels before mating, numbers of pups, degree of prolactin receptor deletion in the transgenic mouse lines). Most likely, we believe this might have been some non-specific variability introduced by the animals’ state of stress or anxiety on the day of measurement. The loss of the early pregnany effect in the 3 different KO groups was also remarkably consistent. Indeed, in our experience, the effect of the Prlr KO on RWA activity was more pronounced than that seen for maternal behaviour. Both the CKC and vGat-induced deletion of Prlr expressed relatively normal maternal behaviour, presumably because the remaining Prlr expression in the MPOA (approx. 50% of neurons) was sufficient to express this behaviour. It was only when we knocked out Prlr in all neurons in the MPOA (using the AAV-Cre) did we see failure of maternal behaviour (as reported previously, and replicated here). This may mean that it is the other, non-vGat neurons in the MPOA that are more important for maternal behaviour, or that these circuits can compensate for a partial loss in Prlr-mediate inputs. In contrast, the pregnancy-induced reduction in RWA was lost to a significant degree in all 3 different Prlr KO groups. We have added this to the discussion (lines 503-509)

3) The reviewers agree that a strong connection between prolactin action and GABA neurons specifically in the POA was not made and that those claims need to be tempered or bolstered by additional experiments. Even though several mouse lineages were used, the authors did not provide a conclusive model that explain how prolactin action on GABAergic neurons of the preoptic area regulates the voluntary physical activity during pregnancy. In the Discussion, they postulate a possible regulation of dopamine secretion and consequently the rewarding aspect of the wheel running, but more experiments in this aspect are necessary to support this claim or these comments need to be clarified as being speculative. There are many additional questions that could be answered or speculated upon: Is the prolactin effect stimulatory or inhibitory in these cells? Is this a mono- or polysynaptic circuit that controls VTA dopamine neurons? How prolactin interacts with estrogen receptor α in the MPOA (the authors provided reference that ER-α cells in this area regulates wheel running)? How the neurocircuit that controls maternal behavior is related to the neurocircuit that controls voluntary activity (is there a overlap or they are different neurons)? Thus, additional experiments to describe the exact mechanism involved in the prolactin`s effects on voluntary activity seems to necessary otherwise this paper is mostly observational and it is actually not quite surprising that preoptic area mediates behavioral aspects controlled by prolactin (e.g., prolactin effects on maternal behavior is mediated by this area, so other behavioral modifications during the pregnancy/lactation cycle may also be part of these same neurocircuit).

We acknowledge that many of the questions raised in this comment remain unanswered, and we have tried to ensure that our discussion of the current set of data has been appropriately tempered such that it does not overstate the evidence. We have focused on trying to narrow down the site of action of prolactin in mediating this behavioural response. We accept that in hindsight, the role of the MPOA in mediating these responses is perhaps not surprising, because prolactin exerts such significant effects on maternal behaviours through this region, and because single cell RNASeq studies have confirmed that it contains such a heterogeneous range of neuronal populations that express the Prlr (PMID:30385464). But at the outset of these experiments, we did not predict this. While a few studies have linked the MPOA to RWA, most studies looking at RWA focused on other circuits, such a lateral hypothalamus inputs to the VTA (PMID: 26341832, PMID: 30010830, PMID:30069057), or the evidence showing a role for arcuate kisspeptin neurons in influencing RWA (PMID:30744968). In fact, we were most surprised when we found that deletion of Prlr on kisspeptin neurons had little effect, as this was our favoured hypothesis *a priori*. We have now tried to incorporate these additional questions into the discussion, and ensured that we are quite clear about what our data shows, and what is speculation about potential mechanism that will require extensive additional experimentation to assess (lines 364-373, 459-473). As discussed in point 2, above, while there might be overlap with the circuits controlled pup-directed maternal behaviour, there are also differences, in that maternal behaviour seems to cope better with Prlr deletion from sub populations (such as vGat containing neurons), whereas the prolactin effect on RWA was largely lost with the sub-population-specific deletion of Prlr.

4) In 2018, the authors published a paper describing reduction in voluntary physical activity in mice [PMID: 29738792]. In that study, the authors showed reduced running wheel activity, like the present study, but differently than here they showed normal spontaneous ambulatory activity inside the cages. In the present study, the authors showed reduced daily ambulation (Figure 1F). The authors did not discuss the possible explanations for these divergent findings and this is important to differentiate the behavior in running wheels (rewarding task) with simple ambulatory movements (possible exploratory behavior).

In the 2018 paper referred to here, mice were housed with running wheels available at all times, and thus, their total physical activity levels included RWA (indeed, it was dominated by RWA). This impacts on their time available for engagement in other ambulatory movements in the cage (for example, we have found that nonpregnant mice with access to wheels will only walk about 100m per day around their cage, whereas when housed without wheels, their ambulatory activity is doubled PMID:29738792). In the new data, to specifically assess ambulatory activity, we housed the mice without access to running wheels, and thus, their physical activity levels were only simple ambulatory movements. In both cases, the total engagement with physical activity was lowered by pregnancy. We have included this information in the text (lines 98-99, 107-112, 148). In the mice with access to running wheels, the reduction in overall activity was only detected as a major reduction in RWA, likely because this accounts for by far the greatest proportion of their overall physical activity levels.

5) Although there is no doubt that deletion of prolactin receptor causes the maintenance of a higher running wheel activity during pregnancy, this behavior is still reduced during mid and late pregnancy. Thus, this behavior is not completely dependent on prolactin action. Although the authors discuss this point, the tone in many parts of the manuscript (title, abstract, results, conclusion) is that this behavior relies completely on prolactin signaling. Thus, it is important to say clearly that although prolactin regulates this behavior, there is likely other factors involved (not only in the Discussion).

We acknowledge this point, and agree that there are important distinctions to be made. It is clear that during mid and late pregnancy other factors also contribute to reducing overall levels of RWA, and we hypothesize that weight gain and physical constraint caused by fetal growth are likely to be involved. It is also possible that other pregnancy-specific biological factors contribute (such as the rising levels of progesterone). We have changed the title and abstract to be clearer on this, and added further data addressing this point, and extensive additional discussion (lines 36-37, 93-95, 249-265, 318-328, 396-397, 416-420, 475-495, 522-526, 535-540). The most obvious role for prolactin is seen in early pregnancy, apparently driving the abrupt decrease in RWA in the first 1-2 days following mating. While it is clear is that some other factors are involved in the progressive further reduction of RWA seen in the 2^nd^ and third trimesters, our data demonstrate that prolactin still plays a role throughout this period, as the KO mice still run more than controls at all timepoints during pregnancy, even when heavily pregnant in the days leading up to parturition. We have also added additional data evaluating RWA during lactation (Figure 7). Here, the physical constraints of pregnancy have resolved, but the suppression of RWA is sustained. Interestingly, our data support a role for prolactin in suppressing RWA in the initial stages of lactation, but this is not sufficient to account for the complete suppression of RWA right through until weaning. We provide some discussion about what those other factors might be. Collectively, the engagement of multiple signals to drive a reduction in RWA in pregnancy and lactation suggests that there must be some biological advantage to successful reproduction of inducing this profound change in behavior. Under the conditions we housed our animals in, there were some consequences of the increased running activity on pregnancy outcome, in terms of pup growth – but there was not any catastrophic failure. Under less optimal conditions, however, a failure to show this adaptive response might have more serious consequences.

6) Continuing the previous comment, the figures emphasize early pregnancy (which is when the authors found the most robust findings), but neglected mid and late pregnancy (these data are not shown in the figures as early pregnancy). Thus, the complete information about the changes in all pregnancy periods are important, especially because it is clear that prolactin receptor knockout mice still present reductions in this behavior, even though in a lesser extend compared to control animals.

We agree with this comment, and have now added figures highlighting RWA in later stages of pregnancy (Figure 4G, H, L, M, Figure 5 E, F, Fig6 F, G, lines 249-265, 318-328, 360-362, 392-395, 475-495). As indicated in the comment, this shows the ongoing contribution of prolactin to the suppression of RWA at these later timepoints. As mentioned above, we have also added a figure evaluating the role of Prlr in lactational changes in RWA (Figure 7, 399-410, 522-526, 535-540), and added additional discussion about this point.

7) Based on the results of this manuscript, I would expect estrous-cycle associated prolactin fluctuations to impact WRA. Given this, it is surprising that WRA is increased on proestrus, when circulating prolactin levels are high. Can the authors comment on this? Related: how do circulating levels of prolactin compare on proestrus versus early pregnancy in mice?

This is a fascinating question, and one that we have also considered. In fact, the increase in RWA during the estrous cycle is detected during the night leading up to proestrus (i.e. is measured on proestrus, but represents what has happened in the 24 hours prior to proestrus. RWA is then reduced over the next 24 hours leading up into estrus (including the time of ovulation and the expression of reproductive behaviour)). There is actually very little data showing high circulating prolactin levels on proestrus in mice. While one early study (Michael 1976, PMID: 1033549) found increased prolactin at the start of the dark phase on proestrus in a mixed bred strain, others using this same strain of mouse only detected a much smaller increase in prolactin concentrations that did not occur until after the start of the dark phase (Deleon et al., 1990 PMID: 2374113). In other studies, no significant rise in prolactin was detected on proestrus and any mild elevation was seen in the middle of the dark phase (Yani and Nalasawa 1974, Sinha et al., 1976 PMID: 1237395). In particular, Shinha et al., 1976 (PMID: 1237395) measured prolactin levels in the afternoon of proestrus in C57BL6 mice and found no significant increase. Because of this lack of information, we have recently completed a study evaluating serum prolactin during the estrous cycle in mice (Phillipps et al., in preparation). The bottom line is that we can detect a small but significant rise in prolactin on proestrus, compared with diestrus. While it does peak in the afternoon of proestrus, around the time of the preovulatory LH surge, it is not as pronounced and well-defined as the proestrous prolactin surge seen in rats. Importantly for the current discussion, however, this rise in prolactin is occurring at a time when RWA is declining, after a peak on the night leading up to proestrus. Our previous work (Larsen et al., PMID: 20484459) has shown robust mating-induced surges in prolactin secretion from the afternoon of day 1 of pregnancy, before the start of the dark phase, which would be the timing when increased endogenous prolactin may have the most influence on peak running wheel activity.

8) Throughout the manuscript the authors use the term "virgin" to refer to non-pregnant animals. Virginity is a state of never having copulated and can be entirely independent of pregnancy and so is not the correct descriptor here. Moreover, it's use suggests that the reduction in locomotion only occurs in the first pregnancy following a "loss" of virginity. Please replace this term with either "non-pregnant", or "cycling". It also raises the question, is the same degree of reduction in RWA seen in multiparous females? Or is this phenomenon limited to the first pregnancy, which would change the interpretation considerably.

This point is well-made, and we have modified the text as suggested. For accuracy, we have adopted the term: “non-pregnant, non lactating” as the description for these animals, and having defined that, more typically use the term “non-pregnant” in referring to this group. Although we like the term “cycling”, we have found that this is confusing for many researchers who are not routinely thinking about the estrous cycle of mice (particularly in a paper about running!). The question regarding what happens with multiparity is an excellent one. At this stage, we do not know, but the fact that “non pregnant” levels of RWA activity return immediately upon weaning of pups would suggest that this is a transient effect specific to pregnancy and lactation, and would be likely to be repeated in subsequent pregnancies.

9) Please include full statistical reporting (e.g., F, t statistics, degrees of freedom etc.) in the main manuscript.

This has been done, as requested.